



# A satellite-derived database for stand-replacing windthrows in boreal forests of the European Russia in 1986–2017

Andrey N. Shikhov[1], Alexander V. Chernokulsky[2], Igor O. Azhigov[1] and Anastasia V. Semakina[1]

[1]Perm State University, Perm, 614990, Russia
[2]A.M. Obukhov Institute of Atmospheric Physics, Russian Academy of Sciences, Moscow, 119017, Russia

*Correspondence to*: Andrey N. Shikhov (shikhovan@gmail.com)

**Abstract.** Severe winds are among the main causes of natural disturbances in boreal and temperate forests. Here, we present a new GIS database of stand-replacing windthrows in the forest zone of the European Russia (ER) for the 1986–2017 period. Delineation of windthrows was based on the full archive of Landsat images and two Landsat-derived products on forest cover change, namely the Global Forest Change and the Eastern' Europe Forest Cover Change datasets. Subsequent verification and analysis of each windthrow was carried out to determine a type of related storm event, its date or date range,

and geometrical characteristics. The database contains 102,747 elementary areas of damaged forest that were combined into 700 windthrows caused by 486 convective or non-convective storm events. The database includes stand-replacing windthrows only, which an area > 5 ha and > 25 ha for events caused by tornadoes and other storms, respectively. Additional information contained weather station reports and event description from media sources is also provided. The total area of windthrows amounts to 2966 km$^2$, that is 0.19% of the forested area of the study region. Convective windstorms contribute

82.5% to total wind-damaged area, while tornadoes and non-convective windstorms are responsible for 12.9% and 4.6% of this area, respectively. Most of windthrows in the ER happen in to summer that is in contrast to Western and Central Europe, where windthrows mainly occur in autumn and winter. The compiled database provides a valuable source of spatial and temporal information on windthrows in the ER and can be successfully used both in forest science and severe storm studies. The database is available at https://doi.org/10.6084/m9.figshare.12073278.v3 (Shikhov et al., 2020).



## 1 Introduction

Forest is a valuable natural resource that is important for economy, society and sustainable development. Forests ecosystems are regularly exposed by natural disturbance agents such as fires, droughts, insect outbreaks, and windstorms. Being an

intrinsic part of forest ecosystem dynamics (Attiwill, 1994; Seidl et al., 2017), natural disturbances cause substantial environmental and economic damage (Schelhaas et al., 2003; Gardiner et al., 2010; van Lierop et al., 2015). In boreal and temperate forests, windstorms consitutes one of the main drivers of natural disturbances (Forzieri et al., 2019). In Europe, windthrows contribute more than a half to the total area of natural disturbances, including abiotic and biotic causes (Schelhaas, 2003; Gardiner et al., 2010).

Recently, disturbance regimes have changed considerably in many forest ecosystems worldwide (Seidl et al., 2011, 2017; Senf et al., 2018). Particularly, occurrence and severity of disturbances both has increased in different regions, including those related to forest fires (Westerling, 2016; Kukavskaya et al., 2016), insect outbreaks (Kautz et al., 2017), droughts (Millar et al., 2015). Researchers have revealed a statistically significant increase of wind-related forest disturbances in the Western, Central, and Northern Europe (Seidl et al., 2014; Gregow et al., 2017), and in the European part of Russia (Potapov

et al., 2015).

The observed increase in the frequency and severity of windthrows is associated with changes in forest structure like a rise of growing stock and median age, primarily in coniferous forests (Schelhaas et al., 2003; Senf et al., 2018), and with climatic changes as well (Overpeck et al., 1990; Lassig and Moĉalov, 2000; Seidl et al., 2011; 2014; 2017). An intensification of winter windstorms (Gardiner et al., 2010; Usbeck et al., 2010; Gregow et al., 2017) and an increase in the frequency and

intensity of severe convective storms in a warm season (Overpeck et al., 1994; Diffenbaugh et al., 2013; Chernokulsky et al., 2017; Radler et al., 2019) can be considered as the main climatic drivers for increasing of wind-related damage in boreal and temperate forests.

For correct attribution of forest windthrows to particular causes, it is important to obtain corresponding data on such events. Recently, several long-term databases of windthrows in boreal and temperate forests, often together with other types of

disturbances, have been collected at a national and macro-regional scale. The longest windthrows data series have been compiled in Sweden (Nilsson et al., 2003) and Switzerland (Usbeck et al., 2010) based on literature reviews and forestry services reports. European Forest Institute presented the database of destructive storms in European forests for 1951−2010 (Gardiner et al., 2010). A new GIS database of wind disturbances in European forests has been compiled in 2019 by aggregating multiple datasets collected by 26 research institutes and forestry services across Europe (Forzieri et al, (2019). It

comprises more than 80.000 forest areas that were disturbed by wind in 2000-2018. Compare to other European countries, windthrows in Russia remain substantially understudied. Long-term databases of windthrows events have been collected only for individual regions, for example, for the Middle Ural (Lassig and Moĉalov, 2000).





The main data sources of exist windthrow databases in Russia were the literature reviews, reports of forestry services, aerial observations and field investigations (Skvortsova et al., 1983; Lassig and Moĉalov, 2000). Meanwhile, satellite images have
become the important data source for windthrows monitoring in Russian forests in recent decades (Krylov et al., 2012). Indeed, satellite data can be especially informative for studying Russian low-populated boreal forests, known in Russia as the taiga, which represent the largest forested region on the Earth. They cover approximately 7.63 million $km^2$, which is 22% of the world's forest areas (WWF Russia's boreal forests, 2007).

Use of satellite images for obtaining information on windthrow was proposed back in 1975 (Sayn-Wittgenstein and
Wightman, 1975). However, the widespread utilising of satellite data to estimate the inter-annual variability of windthrows damage (e.g. Fraser et al., 2005; Baumann et al., 2014) became feasible after the publication of the free-available Landsat archive (Wulder et al., 2012), and two Landsat-based products, namely the Global Forest Change (GFC) map (Hansen et al., 2013) and the Eastern' Europe Forest Cover Change (EEFCC) (Potapov et al., 2015). Thus, GIS databases of windthrows have been collected for some Russian regions based on Landsat archive and GFC data, i.e., for the Ural and north-eastern
part of the ER (Shikhov and Zaripov, 2018; Shikhov et al., 2019), Kostroma region and adjacent areas (Petukhov and Nemchinova, 2014), and South Sakhalin (Korznikov et al., 2019). Shikhov and Chernokulsky (2018) found 110 previously unknown tornado-induced windthrows in the ER based on satellite images. However, for the entire ER, there are only rough estimates of storm-related forest damage (Potapov et al., 2015).

In this study, we present a detailed GIS database of relatively large windthrow events in the forest zone of the ER for the
period 1986-2017. We use the archive of Landsat images, the Landsat-based forest loss data products GFC and EEFCC, high-resolution satellite images from the public map services, supplementary information including weather stations observations, databases on hazardous weather events, damage reports in the media sources, and reanalysis data. We describe the used data and the study region in section 2, and explain the database structure in Section 3. Section 4 describes windthrow delineation process and assessment of the geometrical parameters of windthrows. Section 5 presents spatio-
temporal variability of windthrows and distributions of their geometrical characteristics. Section 5 discusses the main limitations of the method and the compiled dataset, while section 6 draws the main conclusions of the paper.

## 2 Region and data

### 2.1 The study region

The study region includes the forest zone of the ER (Fig. 1) between the forest-steppe transition zone on the south and forest-
tundra transition zone on the north. The availability of the EEFCC dataset determines the eastern boundary of the study region that broadly coincides with the Ural Ridge.

We used the 250-m resolution map of the vegetation cover of Russia (Bartalev et al., 2016) to estimate forest-covered area and dominant forest species (Fig. 1). Forests cover 54.6% of the study regions, while individual forested area within this region is typically > 100 $km^2$. The most widespread dominant forest species are coniferous (*Picea abies, Picea obovata,*



*Pinus sylvestris*), small-leaved (*Betula pendula, Betula pubescens, Pópulus trémula*) and some broadleaved species (*Tília cordáta, Quercus robur et al.*) (Kalyakin et al., 2004). Secondary (re-grown after logging or wildfires) small-leaved and mixed forests cover approximately 61% of the total forested area. Old-growth dark-coniferous forests are widespread on the western slope of the Northern Ural and the adjacent plain, and pine forests cover the large area on the northwest of the ER (Fig. 1).

## 2.2 Initial data

We used multiple data sources to collect information on windthrows for the 1986-2017 period. Particularly, we utilized satellite data to delineate windthrows and determine a storm event type and additional information to determine the dates of storm events.

*Primary information for windthrow delineation and verification*

• The Landsat-based GFC data were utilised to search and delineate windthrows occurred in 2001–2017. The data come as the integer raster with a 30 m cell size. It contains information on stand-replacing forest disturbances that classified with a one-year step. In the boreal forest regions, the overall accuracy of the forest loss detection in the GFC is 99.3%, while user's and producer's accuracies are 93.9% and 88.0%, respectively (Hansen et al., 2013). Here, producer's accuracy is the ratio of correctly classified forest loss area to the actual forest loss area; user's

accuracy is the ratio of correctly classified forest loss area to the same area according to the verified forest loss area. The data were downloaded from http://earthenginepartners.appspot.com/google.com/gMG7KbLG.

• The EEFCC dataset was used to search and delineate windthrows occurred in 1986–2000. The data come as the integer raster with a 30 m cell size. It contains information on forest loss that classified into four broad periods: 1986-1988, 1989-2000, 2001-2006 and 2007-2012. This rough time determination is associated with rareness of the

Landsat images between 1989 and 1998. The detection of gross forest loss in the EEFCC has producer's and user's accuracy of 88% and 89%, respectively (Potapov et al., 2015). The data were downloaded from https://glad.geog.umd.edu/dataset/eastern-europe-forset-cover-dynamics-1985-2012/.

• Landsat images, i.e., images from the Landsat Thematic Mapper (TM), Enhanced Thematic Mapper Plus (ETM+), and Operational Land Imager (OLI), were used to confirm the wind-related nature of forest disturbances, determine

the storm types, dates (or ranges of dates) of windthrows occurrence in 1986–2017. It addition, many windthrows occurred before 2001 were delineated with Landsat images (see section 3.1.3 for details).

• Sentinel-2 images were used to confirm the wind-related nature of forest disturbances, determine the storm types, dates (or ranges of dates) of windthrows occurrence for the 2016-2017 period. The data were downloaded from https://earthexplorer.usgs.gov/

• High-resolution (0.5–2 m) satellite images, hereinafter HRI, were used to discriminate the type of a storm event — windstorm or tornado — causing a windthrow. The HRI are available from 2001 to the present. The HRI images



were downloaded from public map services, i.e., Google Maps, Bing Maps, Here, ESRI Imagery, and Google Earth Pro.

*Additional information on storm events*

• Information of 3-hourly weather reports was used to determine storm event dates, and match the reported wind gusts, if any, with windthrows events. We utilized information on observed wind speed, precipitation, hail and thunderstorm occurrence. The routine meteorological observations have been collected at 402 meteorological stations located within the studied area and have been initially processed at the All-Russian Research Institute of Hydrometeorological Information—World Data Center (RIHMI-WDC) from 1966 to the present (Bulygina et al.,
130 2014).

• Monthly reviews of hazardous weather events occurred in Russia, which are published in the Russian Meteorology and Hydrology journal (http://mig-journal.ru/en/archive-eng) but not translated, were also used to determine storm event dates for the 2001-2017 period. Additionally, these reviews contain the descriptions of hazardous weather events and damage reports. We included this information into our database.

• The RIHMI-WDC database of hazardous weather events (Shamin et al., 2019) and information from regional departments of the Russian state weather service were also utilized to determine the dates of several storms that caused windthrows in 1986-2017.

• Media news and witness reports in social networks, including photos and videos, were used for obtaining additional information on the type of event, i.e. tornadic or non-tornadic, for the 1986-2017 period.

• Data from meteorological satellites Terra/Aqua MODIS (from 2001) and Meteosat-8 (from 2016) were used for obtaining additional information on storm events causing windthrow, especially to determine storm date and time. In particular, the Collection 6 MODIS Active Fire data (Giglio et al., 2016) were used to discriminate fire- and wind-related forest disturbances in 2001–2017. Data were downloaded from https://earthdata.nasa.gov/data/near-real-time-data/firms.

• Data from Russian weather radars (Dyaduchenko et al., 2014) were used only for several events occurred in 2012, 2014 and 2016 to determine the time of storm event causing a windthrow.

**3 Structure of the GIS database**

The compiled database of stand-replacing windthrows in the forest zone of the ER in 1986-2017 is publicly available at https://doi.org/10.6084/m9.figshare.12073278.v3 (Shikhov et al., 2020). We divided the spatial and attributive information
on windthrows into three hierarchical levels that correspond to three GIS layers, i.e., three shapefiles (.shp), in the database:

• "Elementary damaged area" (EDA), that is a single-part polygon of wind-damaged forest;

• "Windthrow", that represent a group of closely spaced forest disturbances, i.e., a multipart polygon, associated with one storm event;



- "Storm event track", that is a cluster of windthrows with identical direction and having the same date (or same date range) of occurrence, which were most likely induced by one convective or non-convective storm.


GIS layers have WGS84 geographic coordinate system (EPSG:4326). The key fields *ID* and *storm_ID* associates each damaged area with the features in the datasets of windthrows and storm event tracks respectively using one-to-many relation. The structure of the attribute tables of each shapefile (stored in .dbf files) is presented at Tables 1–3. The determination process of the presented characteristics is described in Section 4 and schematically presented at Fig. 2.

**4. Methods: windthrow delineation and parameters determination**

The process of windthrows identification and attribution to a particular type includes four stages (Fig.2): (1) delineation of a windthrow using the Landsat-based GFS and EEFCC products or time series of Landsat or Sentinel satellite images, (2) subsequent verification of a windthrow using the HRI and determination of the type of a storm event causing a windthrow, (3) estimation of geometrical characteristics of a windthrow, and (4) determination of storm date or range of dates by

utilizing additional information.

**4.1 Delineation of windthrow areas**

**4.1.1 GFC-based delineation (2001-2017)**

We searched through the GFC dataset for forest loss areas that have characteristic windthrow-like signatures. In particular, we looked for windthrows with the shape that elongated along the direction of storm or tornado movement. Wind-related

forest disturbances rarely have quasi-circular/elliptic or regular shapes that are characteristic for fire-related disturbances and logged areas, respectively (Shikhov and Chernokulsky, 2018, Shikhov et al., 2019). Windstorm- or snow/icestorm-caused windthrows have amorphous spatial structure and a varying degree of forest damage, whereas tornado-induced windthrows have quasi-linear spatial structure and almost total removal of a canopy (Chernokulsky and Shikhov, 2018). After selecting an area affected by a windthrow, we extracted respective pixels from the GFC data and converted them from raster to

multipart vector polygons, which consist of many singlepart polygons, so-called 'elementary damaged area' (EDA). In addition, we removed all EDAs with an area $\leq 0.0018$ km$^2$, that is two GFC pixels. We filtered out such small-scale disturbances since it is impossible to confirm their wind-related origin. Moreover, their area can be almost three times overestimated by Landsat images (Koroleva and Ershov, 2012).

In total, we delineated 450 windthrows using the GFC dataset, and clarified contours of 126 of them manually using the

Landsat, Sentinel-2, or HRI images (see Section 4.2 for details).

**4.1.2 EEFCC-based delineation (1986–2000)**

For the EEFCC data, we performed similarly to the GFC searching and delineation of windthrows with however some limitations. The main limitation is related to the classification of forest losses into broad periods, i.e., 1986-1988 and 1989-





2000. Thereby, windthrow area can be correctly delineated only if it lacks overlap with other forest disturbances, namely

loggings and wildfires, occurred in the same period. For instance, in high-populated areas, salvage loggings are usually performed in 1–2 years for most of wind-damaged forests. Such windthrows were delineated by the Landsat images with semi-automated NDII-based method (see Section 4.1.3). Based on the EEFCC, we were able to delineate windthrows with high confidence mainly in the low-populated northern part of the ER (Fig. 3).

In total, we delineated 153 windthrows using the EEFCC dataset. Contours of the 32% of them were then substantially

clarified manually with the Landsat images, obtained before and after the storm events. Another 22 windthrows that occurred before 2001 were delineated manually using the Landsat images. As for the GFC, we removed all EDAs with an area < 0.0018 km$^2$, since it is impossible to confirm their wind-related origin.

### 4.1.3 NDII-based delineation (1987-2000)

In total, seven large-scale windthrows, occurred before 2001, were delineated by comparing Landsat TM/ETM+ images obtained before and after the storm event in the growing season. We used the difference of Normalized Difference Infrared Index (NDII, Hardisky et al., 1983) to detect and delineate wind-related disturbances. High efficiency of the NDII using for windthrows identification on Landsat images has been shown previously (Wang et al., 2010; Wang and Xu, 2010; Chernokulsky and Shikhov, 1984). NDII was formulated as follows:

$$NDII = (TM4-TM5)/(TM4+TM5), \tag{1}$$

where TM4 and TM5 are the reflectance in the bands 4 (0.85 μm) and 5 (1.65 μm) of Landsat TM/ETM+ data, while the difference was calculate as $\Delta NDII= NDII_{before} - NDII_{after}$, where subscripts 'before' and 'after' denotes two closest to an event cloud-free images obtained, respectively, before and after the windthrow occurrence, but in the growing season only. The masking of forested lands was performed on the 'before' image with the use of Iterative Self-Organizing Data Analysis

Technique Algorithm (Ball and Hall, 1965) unsupervised classification. Then, the NDII was calculated only within the mask of the forested area. The same technique was successfully applied previously to delineate windthrows caused by the 1984 Ivanovo tornado outbreak (Chernokulsky and Shikhov, 1984).

Windthrows and other forest disturbances are characterized by a sharp decrease of the NDII. However, threshold values of ΔNDII for distinguishing between stand-replacing disturbances and moderate damaged or undamaged forests, differ for each

pair of images. In most cases, the pixels with values of ΔNDII that exceed the average value for the forested areas of the entire image by more than two standard deviations, indicate the stand-replacing forest disturbances (Koroleva and Ershov, 2012). However, this value may be less if these disturbances hold the substantial part of the image. We estimated the threshold values from a sample of ΔNDII, obtained within the forested area, and corrected it in several cases to ensure the best fit with the results of the visual identification of windthrows. The threshold values ranged from 1.5 to 2 standard

deviations for different pair of images. On the next step, windthrows were separated from logged areas and other



disturbances (see Section 3.2). The EDAs ≤ 0.0018 km$^2$ were removed. Figure 4 presents the example of the NDII-based identification of the aftermath on 21 June 1998 Moscow windstorm (Los Angeles Times, 1998).

### 4.1.4 Combining delineated polygons to a windthrow and windthrows to a storm

In general, a group of closely spaced EDAs, caused by one storm event, was assigned to one windthrow. By the 'close distance' we meant in most cases a distance of tens or hundreds of meters between the nearest EDAs. However, it may reach 5-10 km, if a windthrow crossed treeless areas. Most of windthrows were extracted from the GFC dataset (450 windthrows), EEFCC dataset (153 windthrows) or with NDII-based methods (7 windthrows). For these windthrows, we first automatically delineated a gross outline of a windthrow as a multi-part polygon and then specified exact contours of its components —

single-part polygons (EDAs); after that, we correctly merged them to a windthrow itself. We delineated other 90 windthrows manually using the Landsat, Sentinel-2, or HRI images — 30, 17, and 43 windthrows, respectively. In this case, we first delineated EDAs and then merged them into a windthrow.

Many storms induced a series of successive windthrows, which are separated from each other by tens or even hundreds of kilometres of undamaged forests, treeless areas or water bodies. In general, we divided the damaged areas into two separate

windthrows (two records in the dataset), if the gap between them exceeded 10 km. The similar threshold value (8 km) was previously used to separate one skipping tornado from two successive tornadoes (Shikhov and Chernokulsky, 2018). A few exceptions were associated with transformations of one windthrow type to another identified by the HRI, i.e., the tornado-induced to non-tornado-induced, and with abrupt change of forest damage degree — from 60-80% to 5-10% of stand-replacing disturbances. In these cases, the distance between two distinct windthrows was less — for instance, the minimum

distance was about 1 km when a tornado-induced windthrow transformed to a squall-induced one.

If several successive or quasi-parallel windthrows have similar direction, differing by no more than 30°, and the same date (or date range) of occurrence, we assigned them to one storm event. The process of windthrow combining to a storm event was based as well on various additional information including the storm dates and types (see next sections), information from weather station reports, eye-witness and newspaper reports, data from meteorological satellites, and so on. In total, the

dataset of storm event tracks contains 486 items.

### 4.2 Verification of windthrows and determination of its type

At the second stage, we performed expert-based verification for each forest disturbance using the HRI or, in the lack of the HRI, the Landsat/Sentinel-2 images. This verification was performed to ensure the forest disturbance was caused by wind

and to determine a type of a storm caused a windthrow. In total, we verified 54% of windthrows with the HRI, mainly for the 2001-2015 period. Other windthrows were verified using the Landsat images (22% of windthrows), the Sentinel-2 images (9%) and additional data sources like weather station and eye-witness reports (15%).

In addition, we used the last cloud-free Landsat or Sentinel-2 image obtained before a storm and first image obtained after to separate windthrows from other disturbances, mainly from logged areas. We removed forest disturbances that were not



related to a storm event (Fig. 5). During the verification, we also found and delineated several storm-damaged areas that were missed in the GFC/EEFCC data. Such areas are located mainly in small-leaved or broadleaved forests. After the verification, we determined the type of a windthrow depending on a weather phenomenon induced this windthrow. We selected tornado-induced and non-tornado-induced windthrows, the latter were divided into induced by convective and by non-convective storms. In turn, non-convective storms include also snowstorms, which are indicated in the database but not

analyzed separately further in the paper. By convective storms we mean squalls and downbursts; however, this more detailed division lacks in the database.

To distinguish tornado-induced windthrows from other wind-related disturbances, we determined the direction of fallen trees using the HRI. Indeed, the main signature of tornado-induced windthrows is the counterclockwise, or infrequently clockwise, rotation of the fallen trees (Beck and Dotzek, 2010; Shikhov and Chernokulsky, 2018). In the lack of the HRI, we

considered three additional signatures of tornado-induced windthrows, namely (1) quasi-linear structure of a windthrow with a ratio of length and width ≥ 10:1, (2) a gradual turn of a storm track, and (3) prevalently total removal of forest stands. Based on these signatures and additional information from weather station reports, witness reports, photos and videos, we assigned the high or medium degree of certainty of storm type determination for each windthrow (Table 4).

Windthrows, caused by non-convective windstorms or snowstorms, have as well specific geometrical features that seen at

satellite images. Specifically, windthrows related to non-convective windstorms typically have enormous length and width of the damage track, up to 200 and 45 km respectively, with however slightly or moderate damaged forests. Caused by non-convective windstorms stand-replacing disturbances are usually occur in dark coniferous forests only (Dobbertin et al., 2002; Schmoeckel and Kottmeier, 2008). Since non-convective storms affect large areas and last for relatively long period, they are typically well-reported by weather stations, which simplify the attribution of related windthrows. In its turn, snowstorm-

induced windthrows are distinguishable from other disturbances primarily based on the dates of occurrence — they happen usually in autumn; although, one severe snowstorm occurred in early summer. It is of note, that we found none of snowstorm-induced stand-replacing windthrow happen in winter.

After the determining of a storm event type, we excluded from the database the tornado-induced windthrows with an area ≤ 0.05 km$^2$ and non-tornado-induced windthrows with an area ≤ 0.25 km$^2$. We took into account the following reasons during

exclusion of such small-scale windthrows:

1. Difficulty to prove that these disturbances are actually were caused by wind, especially in the lack of the HRI.
2. Difficulty to determine wind event dates with the Landsat images for these windthrows.
3. High uncertainty of estimated geometrical characteristics of small-scale windthrows (Koroleva and Ershov, 2012; Shikhov and Chernokulsky, 2018).

Only five squall-induced windthrows with an area < 0.25 km$^2$ were stored in the database, since they are associated with severe weather outbreaks with proven dates. It is of note, that a typical tornado-induced windthrow consist of a relatively small number of EDAs with total removal of forest stands that are well-detected by the Landsat images. In its turn, a typical non-tornado-induced windthrow include larger number of small-scale (i.e., 2-4 Landsat pixels) areas of stand-replacing





disturbances, that are worse detected by satellite images. This difference results in the necessity of using two distinct

thresholds for tornado- and non-tornado-induced windthrows.

## 4.3 Estimation of geometrical parameters of windthrows and its accuracy

We used Landsat data and the Landsat-based products GFC and EEFCC to estimate geometrical parameters of windthrows. We determined the path length ($L$), mean and maximum widths ($W_{mean}$ and $W_{max}$), and damaged area ($A$) for each windthrow using the technique that had been successfully implemented for tornado-induced windthrows (Shikhov, Chernokulsky,

2018). The calculation of these parameters was performed in the Lambert Equal Area and Equidistant projection for North Asia to avoid possible projection-related distortions.

We calculated $A$ in the ArcGIS 10.4 as the sum of area of forest damaged plots, which are attributed to one windthrow. We determined $L$ as a length of the central line drawn through a damaged area, i.e. distance between two farthest points of a windthrow. We calculated $W_{mean}$ as the mean length of several transects that are perpendicular to a storm track with a 200 m

step; this step had been found optimal in terms of quality and counting efficiency (Shikhov, Chernokulsky, 2018). Only stand-replacing windthrows were taken into account in this calculation. In comparison to (Shikhov, Chernokulsky, 2018), where $W_{max}$ were calculated manually using the HRI data, in this study, we assigned the length of the largest transect to $W_{max}$ because the lack of the HRI for many windthrows.

In addition to windthrow characteristics, we estimated geometric characteristics of EDAs and those of storm tracks.

Particularly, for EDAs, we calculated their area $A_{EDA}$. For storm tracks, we estimated maximum and mean width ($W_{TRmean}$ and $W_{TRmax}$), path length ($L_{TR}$), and damaged area ($A_{TR}$). We calculated $W_{TRmean}$ based on the same transects that were used to calculate $W_{mean}$ but without excluding undamaged forests and treeless areas. Similarly, length of the largest transect that includes undamaged forests and treeless areas was assigned to $W_{TRmax}$ (Fig. 6). If a track consists of two (or more) parallel windthrows, then its width was calculated within the outermost boundaries of these windthrows. The same calculation was

performed for $L_{TR}$ in case of two (or more) subsequent windthrows.

We assessed the accuracy of GFC-based estimates of windthrow geometrical parameters by comparing them with the same parameters calculated manually with the HRI using. We performed such procedure for ten windthrows caused by squalls, whose area ranges from 0.26 to 6.09 km$^2$ (Table 5). Distribution of their $A$ is close to the one for the full dataset.

We delineated manually all EDAs within these ten windthrows using the HRI. In total, we found 837 and 947 EDAs,

according to the GFC and the HRI data respectively. Owing to relatively correct georeference of the Landsat data (Landsat Collection 1, 2019), we found no systematic spatial bias between contours of GFC-based and HRI-based windthrows. Despite their general matching, there is no complete overlap due to different spatial resolution of the GFS and HRI (Fig. 7). For example, one GFC-based EDA may intersect with several HRI-based ones, and vice versa. We found, that only 66.5% of the total area is attributed to windthrows in both GFC and HRI, while EDAs with small area can be missed. In particular, 263

HRI-based EDAs with the total area of 0.97 km$^2$ were completely missed in the GFC, while 146 GFC-based EDAs with the total area of 0.52 km$^2$ were missed in the HRI. For overlapped EDAs, we found the mean absolute error and root mean





square error of $A_{\text{EDA}}$ estimates amounted to 27.6% and 13.1%, respectively. We found that the relative error decreases for large EDAs and for those having a simple shape, i.e., quasi-circular. The user's and producer's accuracies increase from 20–25% for EDAs with $A_{\text{EDA}} < 0.01$ km$^2$ to 70–75% for EDAs with $A_{\text{EDA}} > 0.1$ km$^2$. In general, for the overlapped EDAs, the

GFC overestimates their $A_{\text{EDA}}$ (by 4% on average) primarily in coniferous forests. Mutual effect of more frequent omission of small EDAs in the GFC compare to the HRI and overestimation of overlapped EDAs results in approximate equality of total area of delineated windthrows — 17.11 km$^2$ and 17.13 km$^2$ based on the GFC and HRI, respectively.

For entire windthrows, we as well calculated an accuracy of their geometrical characteristics estimating. In particular, we calculated the user's and producer's accuracies of the GFC-based delineation for each of ten selected windthrow. These

accuracies are mainly determined by the complexity of windthrow shapes and composition. In particular, the accuracy is higher for a windthrow consisting of relatively small number of simple-shape EDAs. Otherwise, the accuracy decreases down to 50% for a windthrow with is very amorphous spatial structure. In our sample, the GFC data tends to overestimate the area of windthrows — eight cases out of ten were overestimated. The mean absolute percent error (MAPE) for $A$ is 14.6%. The major overestimation of $A$ by the GFC data, as well as $W_{\text{mean}}$ and $W_{\text{max}}$, was revealed for relatively small

windthrows. This is in line with the previous findings by Koroleva and Ershov (2012) who showed that the reliable estimate (with 15% accuracy) of the damaged area using the Landsat images is possible only for windthrows exceeding 0.026 km$^2$. It is of note, that for tornado-induced windthrows, Shikhov and Chernokulsky (2018) found, that the GFC data generally tends to underestimate $A$, with MAPE amounted to 17.9%.

The assessment of geometrical parameters of windthrows occurred before 2000 and found by the EEFCC is challenging due

to the low availability of the HRI or other independent data sources, e.g. the data of forestry services. Windthrows occurred >20 years ago can be delineated by the HRI only if they passed through old-growth forests that have not been affected by other disturbances, i.e., timber harvesting or wildfires, in subsequent years. Such forests are widespread only in the northeastern part of the ER (Pakhuchiy, 1997). We found five EEFCC-based windthrows occurred between 1998 and 2000 that were most well-detected by the HRI — four tornado-induced and one non-tornado induced. We delineated them with the

EEFCC and the HRI and compare their characteristics (Table 6). We found general overestimation of $A$, $W_{\text{mean}}$ and $W_{\text{max}}$ in the EEFCC, that was larger than in the GFC. It may be related to the inclusion into a windthrow not only real wind-damaged pixels but also surrounding pixels where tree had died after a windthrow appearance mostly because of bark beetles (Köster et al., 2009). Intensity of this mortality is highest at a second year after a storm event (Köster et al., 2009).

## 4.4 Determination of windthrow dates

We aimed to establish the exact date or even the exact time for each windthrow appearance. However, due to data constraints, dates of some windthrows were determined with accuracy < 6 months. We iteratively refined date, or a date range, by using different data. The process, related to the determination of date of tornado-induced windthrows only, had been described previously in (Shikhov and Chernokulsky, 2018).





First, the year of a windthrow can be obtained directly from the Landsat products but with some limitations. In the GFC,
forest disturbances are accompanied with information on the year of event occurrence. However, the exact year is determined correctly only for 75.2% of events; for 21.5% of events, the date can be either a year earlier or a year later (Hansen et al., 2013). In the EEFCC, a year of windthrow occurrence is not explicitly determined and came within the ranges 1986–1988 and 1989–2000 years.

Next, we refined a range of dates based on all available images from the Landsat and Sentinel-2 satellites. The accuracy of
such refinements depends on a frequency of observations and cloudiness. The lowest frequency of satellite observations in the study area, namely 2-4 cloud-free images per year, took place in 2003–2006, 2008, and 2012 years when only Landsat-7 data were available (Potapov et al., 2015). In turn, the highest frequency of satellite imagery, namely ten images per month for a location, was achieved in 2016–2017 after the start of the Sentinel-2 mission.

Further, given the satellite-derived range of event possible dates, we made the subsequent analysis using additional data such
as weather station observations, various databases and reviews on hazardous weather events, damage reports, photos and videos in the media and social networks, and reanalysis data (see (Shikhov and Chernokulsky, 2018) for details). This analysis allowed to establish the exact dates for 48.4% of all windthrows including 39.2% and 59.7% of tornado- and non-tornado-induced windthrows, respectively.

The dates of storm-induced windthrows were defined more successful than those for tornado-induced windthrows due to the
local nature of convective storms, especially of tornadoes, and a relatively large distance between Russian weather stations. Specifically, the average and median distance between nearest weather stations within the study area amounted to 53.7 and 49.9 km, respectively. Wherein, many storm events were reported by weather stations located on a storm path at a distance of 50-100 km from a windthrow, while the closest stations did not reported strong wind gusts since they were away from a storm path. In total, we matched storm reports of weather stations, namely reports with wind gusts ranges from 15 m/s to 34
m/s, only with 34.5% of windthrows with known date.

Another reason for more successful determination of dates for large-scale windthrows than for small-scale windthrows, e.g., tornado-induced, is an increase of probability that a corresponding storm passes through a settlement(s) and this is covered in the media. In total, we used media reports, information from regional weather services, witness photos and videos, existed scientific literature (e.g., Dmitrieva and Peskov, 2013; Petukhov and Nemchinova, 2014; Shikhov and Chernokulsky, 2018;
Shikhov et al., 2019) to specify the date and time of 29.7% of windthrows.

Dates and time of some cases (7.8% of all cases) were established using images from meteorological satellites Terra/Aqua MODIS and METEOSAT-8, and Russian weather radar data (Dyaduchenko et al. 2014). However, the routine usage of these data is time-consuming and limited due to some access restrictions. Subsequent clarification of windthrow exact time can be carried out in further studies.



## 5 Results

### 5.1 Windthrow type

The compiled database includes three shapefiles (.shp), corresponding to three hierarchical levels such as elementary damaged areas, windthrows, and storm events. The database includes 102747, 700, and 486 objects for each level, respectively. The total area of the spatial features is equal 2966.1 km$^2$.

The overwhelming majority of found stand-replacing windthrows in the ER, namely 97.4% of windthrows and 95.3% of wind-damaged area, are associated with convective storms and tornadoes (Table 7). More than a half of all windthrows are tornado-induced with however relatively small damaged area (less than 13%). Non-convective storms and snowstorms are responsible for less than 5% of the area of stand-replacing windthrows in the ER. This is somewhat in contrast to Western and Central Europe, where most of windthrows are induced by non-convective wind events, namely winter storms, caused by strong extratropical cyclones (Gardiner et al., 2010; Gregow et al., 2017). Indeed, winter windstorms affects less Eastern Europe compare to Western and Central Europe (Haylock, 2011). In addition, in the ER and Northern Europe, ground is usually frozen during winter and prevents trees from falling because of windstorms (Suvanto et al., 2016).

Among 486 storm events that caused windthrows, 381 yielded only one windthrow (Fig. 9), primarily tornado-induced. The rest 105 storms resulted in a smaller number of windthrows (319) but larger damaged area — 2276.6 km$^2$, namely 76.8% of all damaged area. Most of these storms induced two or three successive or parallel located windthrows, and only 14 of them caused ≥ 5 windthrows. We found maximum of 17 separate windthrows that related to one storm. We found 71 storm events result in two or more successive windthrows, while 12 storm events lead to formation of two or more parallel windthrows, and 22 storm events include a family of both parallel and successive windthrows. The maximum distance between two nearest successive and two parallel windthrows amounts to 150 and 26 km, respectively.

It should be noted, that a single storm may cause both tornado- and non-tornado induced windthrows, e.g. a supercell can lead to formation of a tornado and a rear-flank downdraft (Karstens et al., 2013) both causing forest damage. In total, we found 30 storms that resulted in formation of two types of windthrows.

We managed to match several storm events with reports at weather stations, in particular the database contains 89 such cases. Among these 89 station reports, we found eight reports with wind gusts ≥ 30 m/s, 14 reports with wind gusts 25-29 m/s, and 30 reports with wind gusts 20-24 m/s. This information have been included in the database, and can be used in further studies to estimate the critical wind speed causing windthrows and to analyse the role of other accompanying weather phenomena, e.g. with snow, heavy rainfall, large hail, etc.

### 5.2 Spatial distribution of windthrows

Windthrows occur in the entire forest zone of the ER (Fig. 9). However, the highest density is observed near the 60° N and somewhat coincides with the highest percentage of forest-covered area (see Fig. 1). It is of note, that two windthrows are located north of 66° N and one of them is even north of the Arctic Circle. The dominant direction of both tornado-induced



and other windthrows is SW-NE (Fig.14b), which is in line with the previous studies on tornado climatology in Northern Eurasia (Shikhov and Chernokulsky, 2018; Chernokulsky et al., 2020).

Three areas, where windthrows have affected more than 0.75% of forests, can be highlighted (Fig. 10a). Two of them are
related to the catastrophic storms which occurred on 27 June 2010 and 29 July 2010. In total, these two storms have damaged 1140 km$^2$ of forests, which is 38.4% of the total area of stand-replacing windthrows in the ER in 1986–2017. The third area is located on the western slope of the Northern Ural and coincides with the largest massive of dark-coniferous forests in the ER (Pakhuchiy, 1997). The largest windthrows occurred here in June 1993, July 2012 and October 2016. The latter was induced by snowstorm. The relatively high frequency of windthrows in this region was emphasized previously
(Lassig and Mocalov, 2000; Shikhov and Chernokulsky, 2018; Shikhov et al., 2019). It was hypothesized that it may be related to the combination of several factors, namely widespread old-growth forests, a high precipitation rate, and large soil wetness, which all contribute to the forests wind susceptibility (Dobbertin, 2002).

The highest density of tornado-induced windthrow is found between 59° and 62° N, 48° and 56° E (Fig. 10, b), which is in a good agreement with the previous estimates (Shikhov and Chernokulsky, 2018). However, when the percentage of tornado-
damaged area of the forested area is considered, then the western part of the ER becomes the most affected by tornadoes (Fig. 10b). It is of note, that higher values of so-called convective instability indices are also observed in this region (Taszarek et al., 2018).

The species composition and age of forest stands have substantial influence on the spatial distribution of windthrows (Dobbertin, 2002, Suvanto et al., 2016; Gregow et al., 2017). However, the available data on the forests species composition
for the entire ER (Fig. 1) have too coarse spatial resolution (i.e., 250 m). Correct estimate the relationships between windthrows and forest characteristics can be carried out in future studies at a regional scale.

### 5.3 Temporal variability of windthrows and storm events

We successfully determined the year of occurrence for all windthrows and the month of occurrence for 263 (67.9%) tornado-induced and 224 (71.5%) non-tornado-induced windthrows. We established the dates of occurrence for 339 windthrows,
including 149 tornado-induced (39.2%) and 187 (59.7%) non-tornado-induced windthrows. It is of note, that the dates of most impacted large-scale windthrows with damaged area > 10 km$^2$ were determined for 44 out of 49 cases (90%). Windthrows with known dates have a total area of 2599 km$^2$, i.e., 87.7% of the total wind-damaged area.

The storm-damaged area has a relatively high inter-annual variability (Fig. 11). The largest area of windthrows, i.e. >1200 km$^2$, is found in 2010, when two exceptional storm events were occurred. An extremely high number of tornado-induced
windthrows occurred in 2009 and 2017. Storm events causing windthrows are observed every year and ranges from 2 to 36, with the maximum in 2012 and minimum in 2001. In general, annual number of windthrows and storm events was lower before 2001 when the EEFCC data were used to identify windthrows, and higher after 2001, when the GFC data were utilized. Annual number of windthrows for these periods amount to 12.1 and 30.5, respectively; in its turn, annual number of





storm events amounts to 8.3 and 20.9. This temporal inhomogeneity, related to different initial data used, should be taken
into account when inter-annual variability is analyzed. More details on dataset limitations are provided in Discussion section.
Windthrows occur in the ER from May to October (Fig. 12). No winter windthrows were found. The seasonal maximum of
the number of windthrows is found in June — both for tornadoes and for other storm events. This is in concordance with the
previous estimates on the tornado climatology (Shikhov and Chernokulsky, 2018; Chernokulsky et al., 2020). Maximum
frequency of the occurrence of storm events causing windthrows is also observed in June. Moreover, more than 90% of
storm events with known dates occur in summer. It is important to note, that we failed to establish the month of occurrence
for 127 tornado-induced windthrows and 98 non-tornado induced windthrows, with the total area of 245 km$^2$.

Sometimes, two or more storm events causing windthrows occurred in ER on the same day. In total, we found seven
outbreaks with more than ten windthrows per day. The most remarkable outbreaks occurred on 18 July 2012 when nine
storms resulted in 25 windthrows, and on 7 June 2009 when five storms resulted in 24 windthrows. However, the largest
forest damage is associated with a single storm, namely the long-lived convective storm "Asta" (Suvanto et al., 2016). This
storm has passed over the northwestern part of the ER and Finland on 29 July 2010 and has damaged 639 km$^2$ of forests in
Russia.

We restored the time of occurrence with 6-h accuracy for 216 windthrows — 136 among them using weather station reports
and 80 using other data sources. We found 122 windthrows (56.4%) occurred between 15.00 and 21.00 of local time (LT),
which coincides with the afternoon maximum of the development of deep convection. However, several most impactful
storms, including for instance the 'Asta' storm, occurred around midnight at LT. No windthrows found between 06.00 and
10.00 LT during the morning minimum of the convection diurnal cycle. The similar diurnal cycle was found for tornado
events in the Northern Eurasia (Chernokulsky et al., 2020).

**5.4 Geometrical parameters of windthrows, elementary damaged areas, and storm tracks**

Area of EDAs varies between 0.0018 to 30.9 km$^2$. Most of EDAs are less than 0.01 km$^2$ (Fig. 13a), but their total area is less
than 10%. In turn, 1% of the largest EDAs account for 36.8% of the total area of windthrows. Using Kolmogorov-Smirnov
(K-S) test, we found that at 0.01 significant level we can reject the null hypothesis that two samples of $A_{EDA}$ within each pair
of windthrow types are drawn from the same distribution (at 0.01 level). Because of small sample size of windthrows
induced by non-convective storms, later in the article we will not discuss the results of K-S test to compare distributions of
characteristics of this type with those of other types.

Tornado-induced windthrows contain fewer plots, than other windthrows (Fig. 13b). Particularly, most of tornado-induced
windthrows include 10–25 EDAs, and only 2.5% of them consists of more than 100 EDAs. In contrast, about 43% of non-
tornado induced windthrows includes more than 100 EDAs, while 5.5% of them consists of more than 1000 EDAs. Based on
K-S test, we found that samples of number of EDAs in tornado- and convective storm induced windthrows are from different
distributions.



A relatively small number of severe storm events are responsible for most of the area of windthrows (Fig. 14a). Indeed, the ten most destructive storm events occurred in the ER over 1986-2017 damaged 1758 km$^2$ of forests, namely 59.2% of the total area of windthrows in the database. This peculiarity is less pronounced for tornado-induced windthrows, since their area usually is less than 10 km$^2$. Particularly, ten tornadoes with the largest area damaged 96.6 km$^2$ of forests — 25.5% of the total tornado-damaged area. Thus, the distribution of tornado-damaged area is less skewed to high values, than the distribution of other windthrows. The K-S test shows that samples of $A$ for tornado- and convective storm induced windthrows are from different distributions.

Length of windthrows ranges from 0.8 km to 283.6 km (Fig. 14b). More than 44% of tornado-induced windthrows have path length < 5 km, while path length 5-15 km is most frequent for non-tornado-induced windthrows. Based on K-S test, we found that samples of number of $L$ for tornado- and convective storm induced windthrows are from different distributions. The maximum length of storm track, consisting of several subsequent windthrows, reaches 544 km. This damage track is caused by the storm on 27 June 2010. In addition, another nine storm tracks have a length exceeding 250 km — most of them are among the most destructive in terms of forest-damaged area. Such series of windthrows with an exceptionally long path length were likely caused by derechos, i.e. long-lived mesoscale convective systems producing widespread damaging winds (Johns and Hirt, 1987). A few derecho events occur each year in Europe, and some of them induced catastrophic forest damage (Taszarek et al., 2019). Although, not a single derecho events have been reported previously in Russia. A more detailed further analysis of these storm events should be carried out to confirm their nature.

Most of tornado-induced windthrows have $W_{max}$ and $W_{mean}$ less than 200 m (Fig. 14 c,d). Instead, the distribution of $W_{max}$ of non-tornado induced windthrows shifted toward larger $W_{max}$. In particular, 103 windthrows (32.9%) have $W_{max} > 1000$ m. The K-S test shows that samples of both $W_{max}$ and $W_{mean}$ for tornado- and convective storm induced windthrows are from different distributions. Width of storm tracks is several times higher than the width of windthrows. Moreover, the $W_{TRmax}$ of windthrows caused by non-tornadic storms is several times higher than the $W_{TRmean}$. $W_{TRmax}$ exceeds 30 km for three widest convective storms — two derechos occurred on 27 June 2010 and 29 July 2010, and one non-convective storm occurred on 7-8 August 1987.

## 6 Discussion: method limitations

The presented database likely lacks many windthrows that occurred in ER in 1986–2017. Specifically, since most of windthrows were delineated from the GFC and EEFCC datasets, objects which are initially missed or underestimated in these datasets, could be as well missed in our database. The performed verification with the Landsat images and the HRI allows to reduce these omissions. In particular, we found several windthrows in small-leaved or broadleaved forests that were significantly underestimated in the GFC dataset.

The efficiency of the method depends on the percentage of forest-covered area. The most reliable estimates of wind-damaged area can be obtained for low-populated northern and eastern regions of the ER, where forests cover 70-90% of the



territory (Bartalev et al., 2016) (Fig.1). In the southern part, the probability of the windthrows omission is higher (Shikhov,
Chernokulsky, 2018).

It is possible to miss windthrow if a storm or tornado passed through areas of intensive timber harvesting or agricultural
lands (Shikhov, Chernokulsky, 2018). Salvage logging performed shortly after a storm event also complicates the
identification of windthrows (Baumann et al., 2014). However, in most cases, the time interval between storm passing and
salvage logging in the ER was quite long, i.e., more than a year, except for more populated southern regions.

Currently, the proposed method requires expert verification at almost all stages, which prevents to switch it into the
automatic mode. The possibility of automated searching throughout the GFC and EEFCC datasets is limited by a wide
variety of geometrical shapes of windthrows and their overlapping with other forest disturbances. The data collection process
requires the use of numerous and diverse sources such as the HRI from various public web-services, weather station reports,
eye-witness and media reports, etc.

While the algorithms for automated forest disturbances detection based on satellite data are well-developed and applied at
the regional-to-global scale (Huo et al., 2019), automated attribution of forest disturbances to their causes, namely
windstorms, logging, wildfires, insect outbreaks, and others, remain a critical challenge for remote sensing-based forest
monitoring. The spectral characteristics of various types of disturbances, e.g., windthrows and logged areas, are often similar
(Baumann et al., 2014) that complicates the attribution automatization. The promising approaches in this process is the
complex use of spectral, temporal, and topography-related metrics (Oeser et al., 2017) as well as implementing of advanced
image classification/segmentation methods (Oeser et al., 2017; Liu et al., 2018; Huo et al., 2019). In future studies, such
approaches can be applied to automate delineation of windthrows in the ER using satellite data of various spatial resolution.

We have to stress temporal inhomogeneity of our database, especially for small-scale windthrows, due to the following
causes:

1. The use of two different Landsat-based products to search windthrow-like disturbances — the EEFCC before 2001
and the GFC after. The GFC data have higher accuracy of forest loss detection and of initial time assigning, than the
EEFCC (see section 4.1 for details), which allows to detect more windthrows. Thus, the annual number of
windthrows 2.5 times higher in the GFC period compare to the EEFCC period.

2. After 2002-2003, the HRI had become available, which made it possible to confirm the tornadic nature of
windthrows. The observed increase in the number of tornado-induced windthrows after 2003 is very likely related
to the appearance of the HRI.

3. The start of the Sentinel-2 mission in 2015 providing the images with a 10 m spatial resolution (Drusch et al., 2012)
had also increased the possibility for windthrow identification.

4. A strong decrease in the volume of timber harvesting occurred in the ER, especially in its northeastern past, after
the Soviet Union dissolution (Potapov et al., 2015). This could led to more omission of windthrows in the late
1980s compare to the subsequent period because of their masking out with logging.

Thus, the presented database should be used for assessing interannual variability with caution. Special assumptions should be made to estimate linear trends. For instance, they can be obtained for particular regions, e.g. for those with little changes of forestry practices, and for relatively large windthrows, that are well-detected from both the EEFCC and the GFC data. For instance, linear trend of number of windthrows with area $\geq 1$ km$^2$ amounts to 0.27 year$^{-1}$ and is statistically significant at 0.05 level[1]. This increase of wind-related windthrows is in line with observed increase of such characteristics as convective precipitation (Ye et al., 2017; Chernokulsky et al., 2019), convective cloudiness (Sun et al., 2001; Chernokulsky et al., 2011), convective instability indices (Riemann-Campe et al., 2009; Chernokulsky et al., 2017) in the ER in the last decades.

**7 Conclusions**

The compiled GIS database contains the most complete information on a relatively large stand-replacing windthrows in the forest zone of the European Russia in 1986-2017. The database contains 102747 elementary damaged areas, combined into 700 windthrows, which were caused by 486 storm events. For each windthrow, we determined its type with degree of certainty, dates or date ranges, and geometrical characteristics. Database also contains weather station reports and links to additional information on storm events from the media. We included into the database only the stand-replacing windthrows with an area $> 0.05$ km$^2$ and $> 0.25$ km$^2$ for the tornado- and non-tornado-induced windthrows, respectively.

The total area of windthrows amounts to 2966 km$^2$, namely 0.19% of the forested area within the study region. Most of windthrows in the ER, i.e., 82.5% of the total wind-damaged area, are related to convective squalls and downbursts, which occur mainly in June and July. The ten most impactful storms are responsible for 59.2% of the total forest damage. More than 55% of windthrows in the database are tornado-induced, but their contribution to total damaged area is much lower — it is less than 13%. Non-convective windstorms and snowstorms caused only 4.6% of storm-damaged area.

The largest area of windthrows is assigned to the 2010 year, when two exceptionally destructive storm events occurred — on 27 June 2010 and 29 July 2010. An extremely high number of tornado-induced windthrows was observed in 2009 and 2017 — 45 and 40 tornadoes, respectively.

The presented method has several limitations which results in spatial and temporal inhomogeneity of the compiled database specifically for small-scale windthrows. Because of influence of forest area percentage and forestry practice, such windthrows can be rather missed in the southern part of the ER compare to the northern part. Because of coarser resolution of the EEFCC data and lack of the HRI, such windthrows can be rather missed before 2001. The obtained increases in number of windthrows and their area are artificial. However, the positive trend is likely real for large-scale windthrows, namely for ones with the area $\geq 1$ km$^2$.

The compiled database provides a valuable source of spatial and temporal information on windthrows in the ER, which previously has been incomplete. On the one hand, the database allows estimate the role of wind-related disturbances in

---

[1] Trends were computed with the Theil–Sen estimator. Significance was obtained with the nonparametric Mann–Kendall test.



comparison with other natural disturbances in forests and improve our understanding of different forest species susceptibility to windstorms. On the other hand, the database presents a unique source of information on storm and tornado events causing windthrows in the low-populated forest zone of the ER. It includes numerous of previously unknown storms and tornadoes,

which caused forest damage, and also clarifies information on known storm events. Thus, the database significantly contributes to the climatology of severe storms and tornadoes in the ER. Based on the compiled database, further studies may be carried out to determine the contribution of climate variability to the inter-annual variability of wind-related forest damage, and to quantify the risk of windthrows in forests of the entire ER.

## 8 Data availability

Data are freely available at https://doi.org/10.6084/m9.figshare.12073278.v3. (Shikhov et al., 2020) and will be periodically updated with new and historical events.

**Author contributions.** ASh and ACh designed the study. ASh and ASe performed windthrows identification using satellite data. ASh and IA carried out an analysis of additional information to determine storm event types and dates. Ash and ACh, with contributions by IA, wrote the initial draft of the paper and produced the maps and figures.

**Competing interests.** The authors declare that they have no conflict of interest.

**Acknowledgements.** The study was funded by the Russian Foundation for Basic Research (projects no. 19-05-00046 and 20-35-70044). The determination of storm track characteristics was supported by the Russian Science Foundation (project no. 18-77-10076).



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





| Field name | Field alias | Type, length | Description |
|---|---|---|---|
| OBJECTID | OBJECTID | Object ID | Index number of EDA |
| ID | Windthrow ID | Short | Windthrow ID |
| Storm_ID | ID of storm event | Short | ID of a storm event |
| Area | Area (km$^2$) | Float | EDA area (km$^2$) |


**Table 1: Attribute table of the GIS layer of elementary damaged areas (EDAs).**




| Field name | Field alias | Type, length | Description |
|---|---|---|---|
| OBJECTID | OBJECTID | Object ID | Index number of windthrow |
| ID | Windthrow ID | Short | A windthrow ID |
| Storm_ID | ID of storm event | Short | ID of storm event |
| Storm_type | Type of storm | String, 10 | A type of a storm that caused the windthrow: convective windstorm, tornado, non-convective windstorm, or snowstorm |
| Certainty | Event certainty degree | String, 20 | The degree of certainty of storm type determination: high or medium |
| Source_1 | Data source for windthrow delineation | String, 50 | Data source for windthrow delineation |
| Source_2 | Data source for windthrow type defining | String, 100 | Data source for windthrow type defining |
| Year | Year | Short integer | The year of the windthrow event |
| Month | Month | Short integer | The month of the windthrow event |
| Date | Storm event date | String, 20 | The date of storm event |
| Date_1 | Date of first image | Date | The date of the last Landsat/Sentinel-2 image that lack the windthrow |
| Date_2 | Date of second image | Date | The date of the first Landsat/Sentinel-2 image, by which the windthrow was detected |
| Time_range | Time range | String, 50 | Time range of storm event (UTC) |
| Time_Src | Data source for determine storm time range | String, 255 | Data source or URL that was used to determine the time range of a storm event |
| N_polygons | Number of single-part polygons | Short | Number of single-part polygons |
| Area | Area (km$^2$) | Float | Windthrow area (km$^2$) |
| Length | Path length (km) | Float | Length of windthrow (km) |
| Mean_width | Mean width of windthrow excluding gaps (m) | Float | Mean width of windthrow (m) — for damaged area only |
| Max_width | Max width of windthrow excluding gaps (m) | Float | Maximum width of windthrow (m) —for damaged area only |
| Mean_w_2 | Mean width of windthrow with gaps (m) | Float | Mean width of windthrow including gaps (m) |
| Max_w_2 | Max width of windthrow with gaps (m) | Float | Maximum width of windthrow including gaps (m) |
| Direction | Direction of windthrow | String, 10 | Elongated direction of windthrow, i.e. direction of storm movement |
| Near_WS | WMO ID of the weather station | Long | WMO ID of the nearest weather station — if the distance between windthrow and weather station is less than 50 km or weather station located on the storm track |
| WS_dist | Distance to weather station | Float | Distance to the nearest weather station (km) |





| | (km) | | |
|---|---|---|---|
| Wind_gust | Wind gust (m/s) | Short | Maximum wind gust that measured by the weather station on a day when windthrow occurred |
| Gust_time | Wind gust time (UTC) | Short | Time of wind gust report (UTC) with 3-hour accuracy |
| Sum_prec | Precipitation amount | Short | Precipitation amount (only for events with heavy rainfall ≥ 30 mm/12h) |
| WS_comment | Additional data from weather station | String, 100 | Additional data on the storm event reported by the weather station, i.e. heavy rainfall (≥ 30 mm/12h), large hail, tornado |
| URL | External URL | String, 100 | URL of the additional data source (newspaper report or video) |

**Table 2: Attribute table of the GIS layer of windthrows in the forest zone of ER (1986-2017).**



| Field name | Field alias | Type, length | Description |
|---|---|---|---|
| OBJECTID | OBJECTID | Object ID | Index number of a storm track |
| Storm_ID | ID of storm event | Short | ID of a storm event |
| Count | Number of windthtows | Short | Number of windthrows caused by a storm event |
| Area_tr | Area (km$^2$) | Float | Total damaged area (km$^2$) |
| Length_tr | Path length (km) | Float | Total path length with gaps, km) |
| Mean_w_tr | Mean width of storm track (m) | Float | Mean width of storm track (km) |
| Max_w_tr | Max width of storm track (m) | Float | Maximum width of storm track (km) |


**Table 3: Attribute table of the GIS layer of storm events tracks.**





| Degree of certainty | Windthrows induced by | | |
| --- | --- | --- | --- |
| | **Tornado** | **Convective storm** | **Non-convective storm** |
| High (>95% likelihood of occurrence) | Independent confirmation of the tornado event (photo, video, etc.); well-detected rotation of the fallen trees (counterclockwise usually); all three additional signatures are confirmed (in the lack of the HRI) | Elongated, but amorphous (mosaic) spatial structure of forest disturbances and a varying degree of forest damage; the direction of the fallen trees generally corresponds to a storm track direction | Independent confirmation of non-convective storm causing windthrow by weather station or/and eye-witness/newspaper report; |
| Medium (50–95% likelihood) | The HRI are unavailable or do not allow to determine the direction of the fallen trees and only two out of three additional signature are confirmed. | HRI are unavailable or do not allow to determine the direction of the fallen trees; quasi-linear structure of a windthrow without turns of a track, and a ratio of length and width < 10:1 | The date of a storm event indicate a low probability of a convective storm (e.g., autumn season) and lack of elongation along the wind direction (especially for windthrows induced by snowstorms) |

**Table 4: The signatures used to assess the degree of certainty of windthrow type determination.**





| Number | Total area (GFC/HRI), km$^2$ | $A$ (overlapped), km$^2$ | Producer's accuracy, % | User's accuracy, % | $L$, km (GFC/HRI) | $W_{mean}$, m (GFC/HRI) | $W_{max}$, m (GFC/HRI) |
|---|---|---|---|---|---|---|---|
| 1 | 6.08/6.49 | 5.04 | 77.6 | 82.8 | 9.4/9.4 | 588/612 | 1433/1467 |
| 2 | 4.36/5.11 | 2.98 | 58.5 | 68.5 | 15.9/17.2 | 290/405 | 860/1798 |
| 3 | 1.74/1.54 | 0.75 | 48.7 | 43.2 | 42.5/42.5 | 104/87 | 542/390 |
| 4 | 1.55/1.31 | 0.79 | 60.3 | 51.3 | 9.0/9.1 | 178/152 | 681/593 |
| 5 | 1.33/0.92 | 0.71 | 77.0 | 53.6 | 6.7/6.8 | 220/145 | 638/510 |
| 6 | 1.00/0.76 | 0.41 | 53.9 | 41.1 | 21.8/21.8 | 86/70 | 343/250 |
| 7 | 0.88/0.76 | 0.41 | 53.9 | 46.6 | 14.6/14.7 | 112/97 | 458/382 |
| 8 | 0.42/0.32 | 0.19 | 59.7 | 44.5 | 7.4/7.2 | 85/53 | 233/179 |
| 9 | 0.27/0.14 | 0.11 | 77.2 | 41.7 | 2.1/2.1 | 136/79 | 306/264 |
| 10 | 0.26/0.25 | 0.15 | 61.4 | 60.0 | 9.4/9.4 | 86/59 | 188/206 |

**Table 5: Comparison of windthrows geometrical parameters estimated using the GFC and the HRI data.**






| Number | $A$, km$^2$ (EEFCC/HRI), | $A$ (overlapped), km$^2$ | Producer's accuracy | User's accuracy | $L$, km (EEFCC/HRI) | $W_{mean}$, m (EEFCC/HRI) | $W_{max}$, m (EEFCC/HRI) |
|--------|------|------|------|------|------|------|------|
| 1 | 3.11/4.18 | 2.58 | 82.96 | 61.72 | 14.6/14.2 | 308/257 | 963/748 |
| 2 | 1.59/2.35 | 1.25 | 78.62 | 53.19 | 16.8/16.9 | 186/148 | 568/491 |
| 3 | 3.48/3.82 | 2.68 | 77.01 | 70.16 | 14.2/14.9 | 305/288 | 1507/1269 |
| 4 | 0.82/1.11 | 0.67 | 81.71 | 60.36 | 10.3/10.4 | 166/158 | 367/332 |
| 5 | 1.09/1.28 | 0.94 | 86.24 | 73.44 | 9.5/10.1 | 171/161 | 380/291 |

**Table 6: Comparison of windthrows geometrical parameters estimated using the EEFCC and the HRI data.**





| Windthrow type | Degree of certainty | Number of windthrows | Damaged area, km$^2$ |
|---|---|---|---|
| **Convective storm induced** | High | 270 | 2371.6 |
| | Medium | 25 | 7.6 |
| **Tornado-induced** | High | 295 | 300.4 |
| | Medium | 92 | 79.2 |
| **Non-convective storm induced** | High | 12 | 131.8 |
| | Medium | 6 | 5.9 |
| **Total** | High | 577 | 2803.8 |
| | Medium | 123 | 92.7 |

**Table 7: Total number of windthrows of different types and corresponding forest damaged area.**



**Figure 1: Land cover types within the study area,** according to the map of vegetation cover of Russia, developed by the Space Research
Institute of the Russian Academy of Sciences (Bartalev et al., 2016).



**Figure 2:** Workflow used for windthrow delineation and attribution.

**Figure 3: Delineation of (a, b) storm- and (c, d) tornado-induced windthrows** based on (a, c) the EEFCC dataset and (b, d) its subsequent verification by the Landsat images, created as a combination of the TM3 (0.66 µm), TM4 (0.85 µm), and TM5 (1.65 µm) spectral bands.



**Figure 4: Windthrow delineation the windstorm occurred on 21 June 1998 in Moscow region based on the NDII difference method:** the Landsat-5 images obtained (a) before and (b) after the storm event — 11 May 1998 and 30 July 1998, respectively; (c) the NDII difference within forest-covered area and (d) the areas with the substantial decrease of NDII.


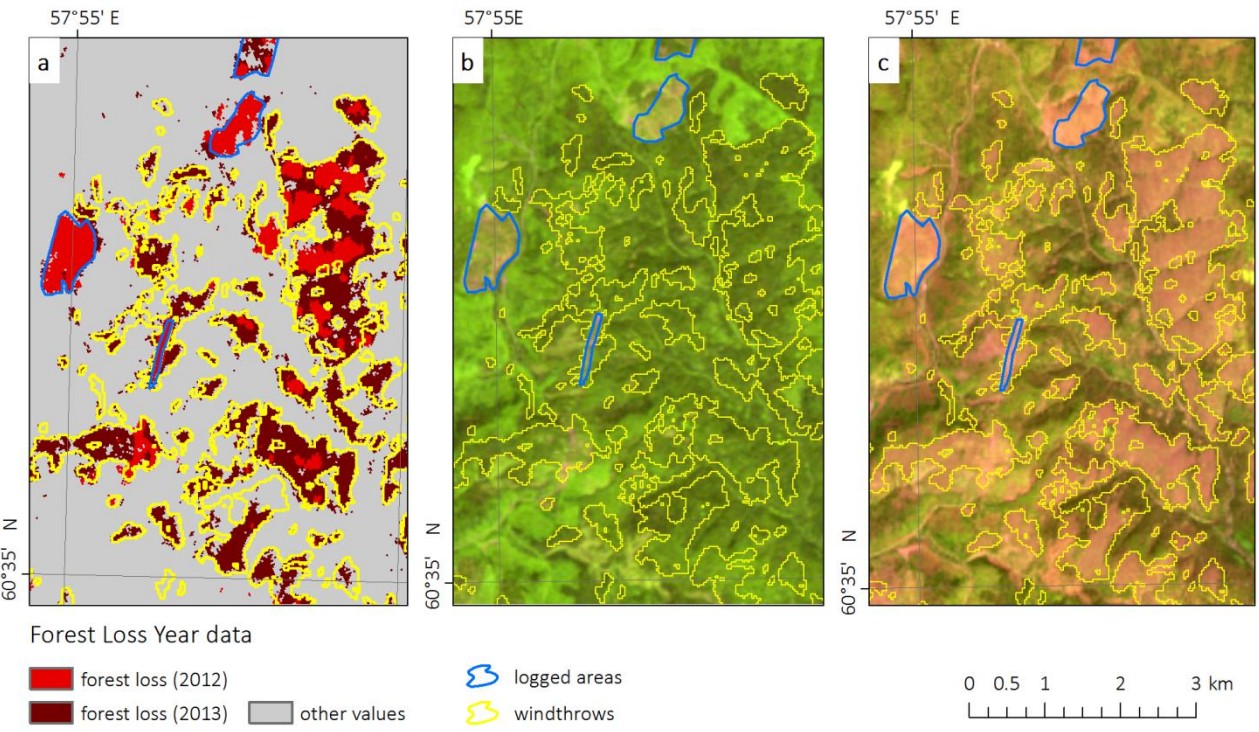


**Figure 5: Separation of the windthrow occurred on 18 July 2012 from logged areas** based on (a) the GFC data on forest losses, and Landsat images obtained (b) before (i.e., 8 July 2012) and (c) after (i.e., 18 Aug 2012) the storm event.

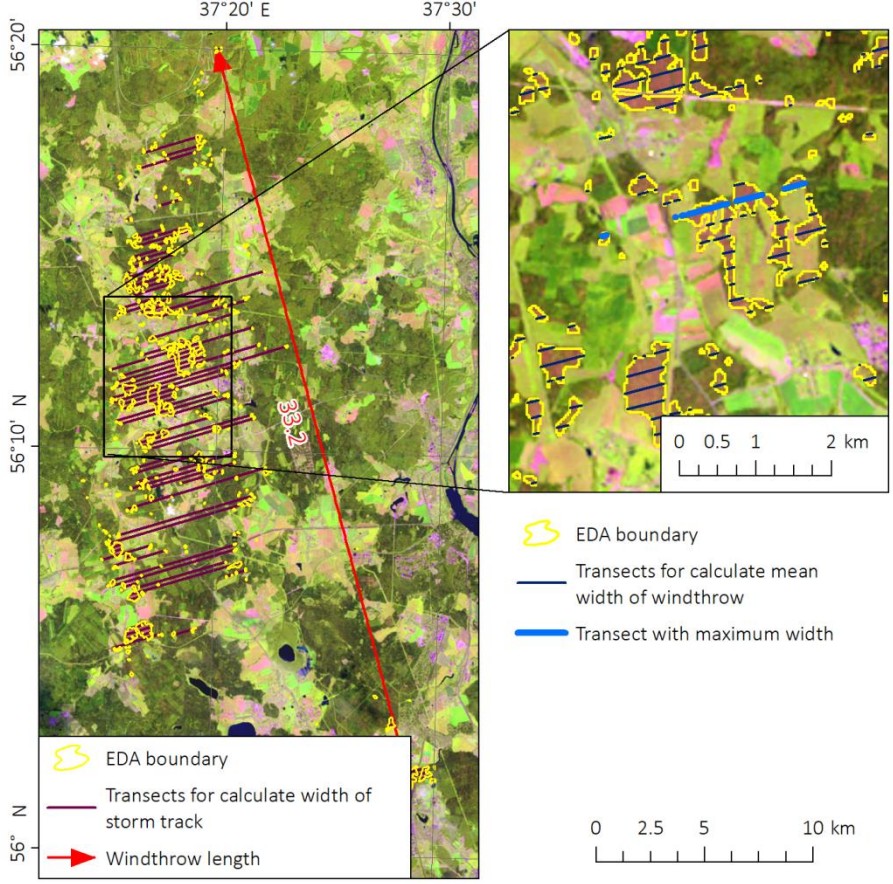


**Figure 6: A scheme for the determination of geometrical parameters of a windthrow based on the Landsat image** using the example of the windthrow in the Moscow region occurred on 21 June 1998.





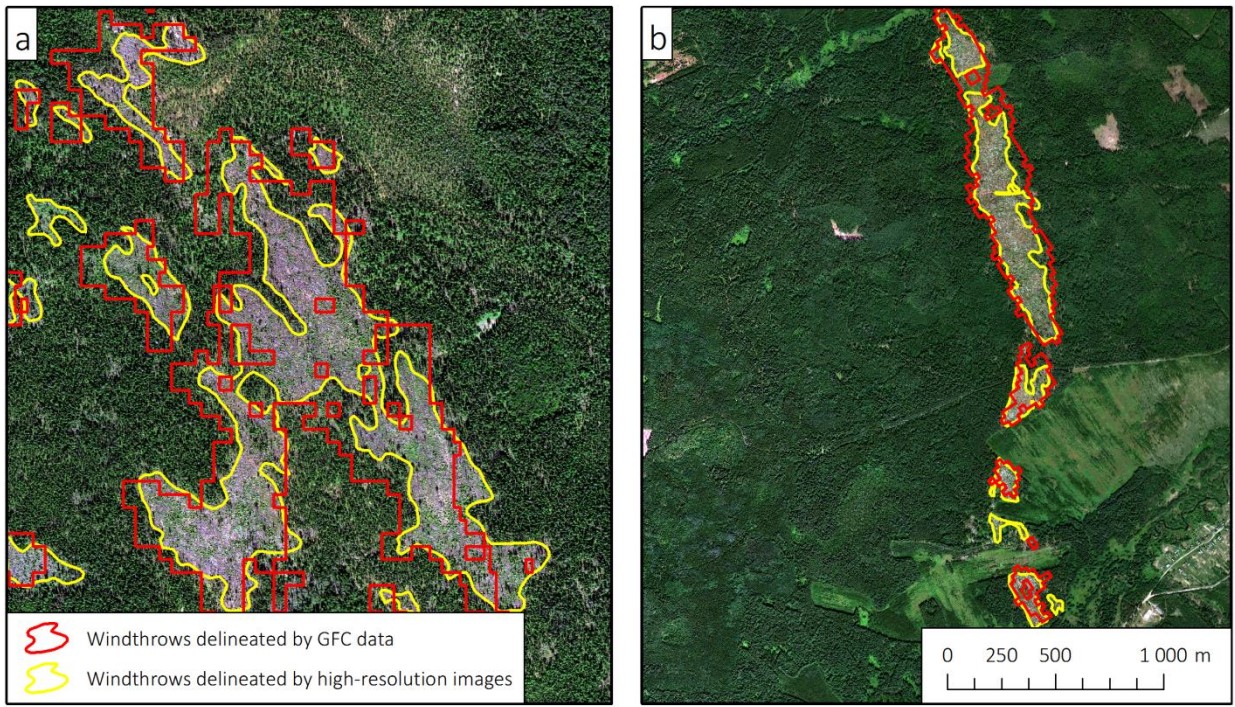

**Figure 7: Overlapping of windthrow areas that extracted from the GFC dataset and delineated manually using the HRI** for (a) **convective-storm** induced windthrow, and (b) tornado-induced windthrow.




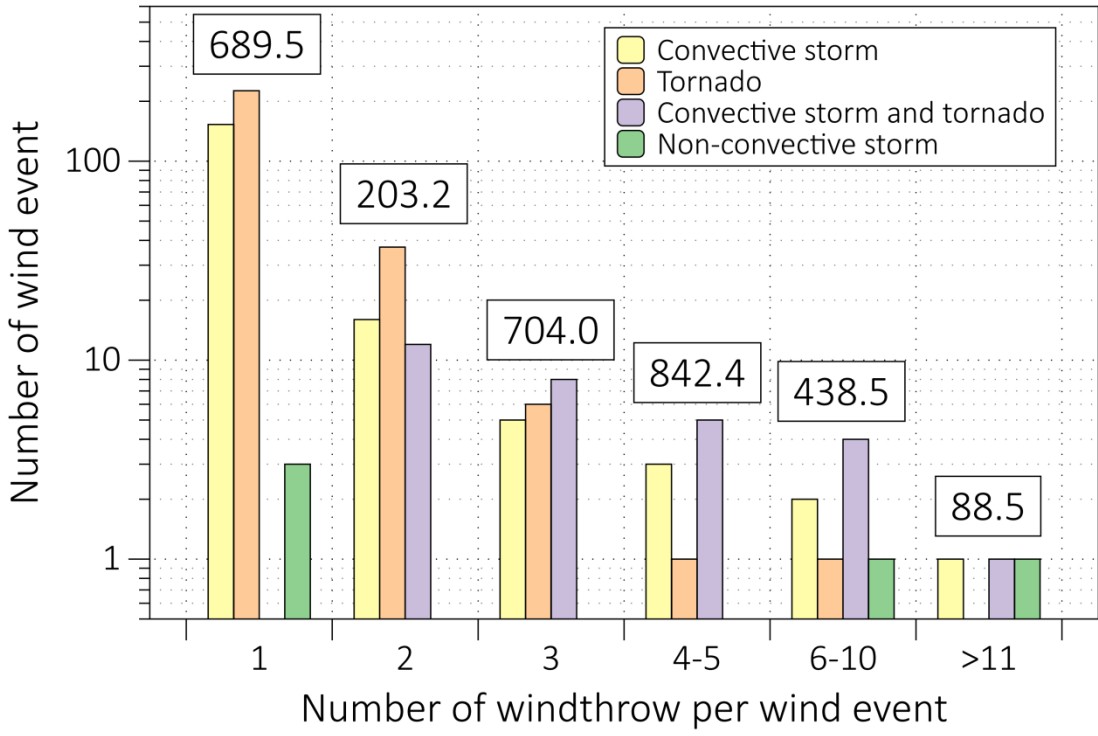

**Figure 8: Number of windthrows per one storm event. Total damaged area (in km$^2$) corresponding to all type of windthrows is shown in box for each category.**




**Figure 9: Spatial distribution of stand-replacing windthrows in the ER in 1986-2017.** The ten most catastrophic windthrows with the largest damaged area are shown by arrows and indicated by the corresponding dates of windthrows. Forest-covered area is estimated according to the data from Bartalev et al. (2016). The inset shows the direction from which windthrows originate.




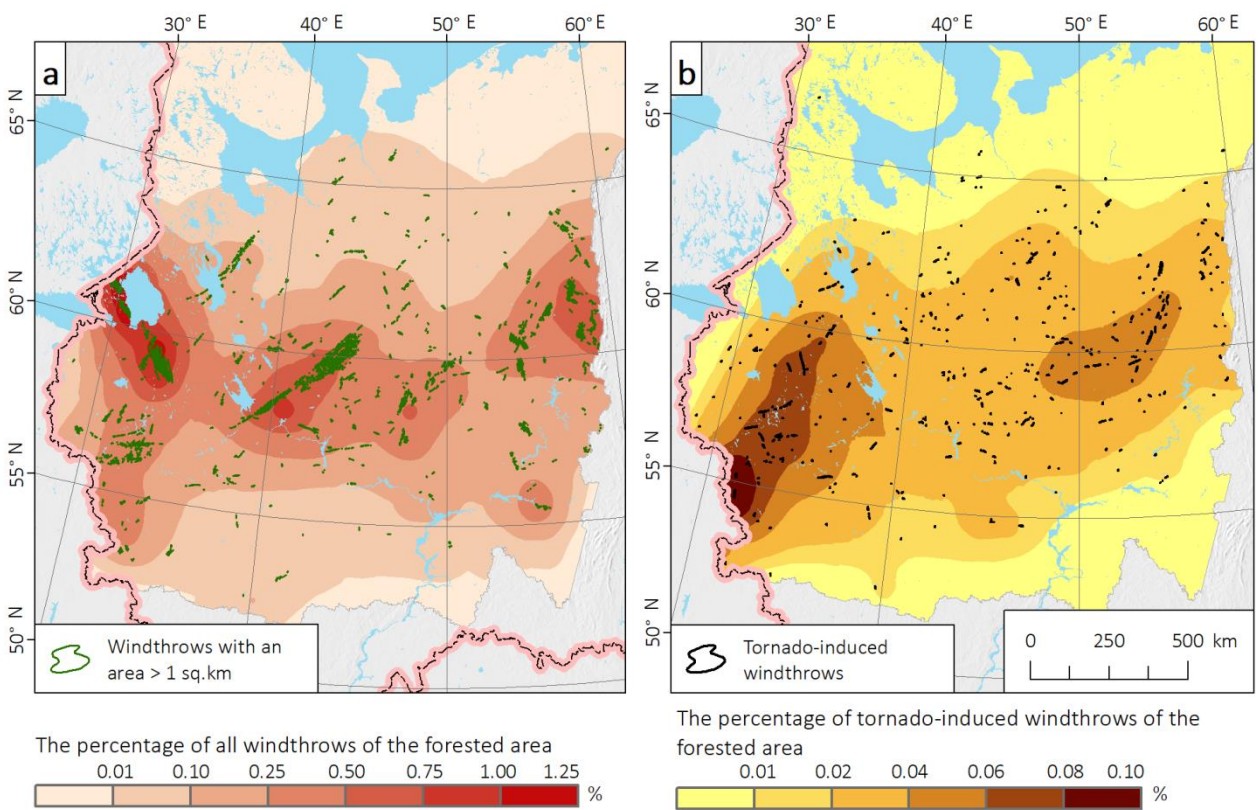

**Figure 10: Ratio of damaged area to the forest-covered area for (a) all windthrows and (b) tornado-induced windthrows only.** The ratio of windthrows area to the forest-covered area was calculated for 100 km² cell and then interpolated with local polynomial interpolation method in the ArcGis Geostatistical Analyst.


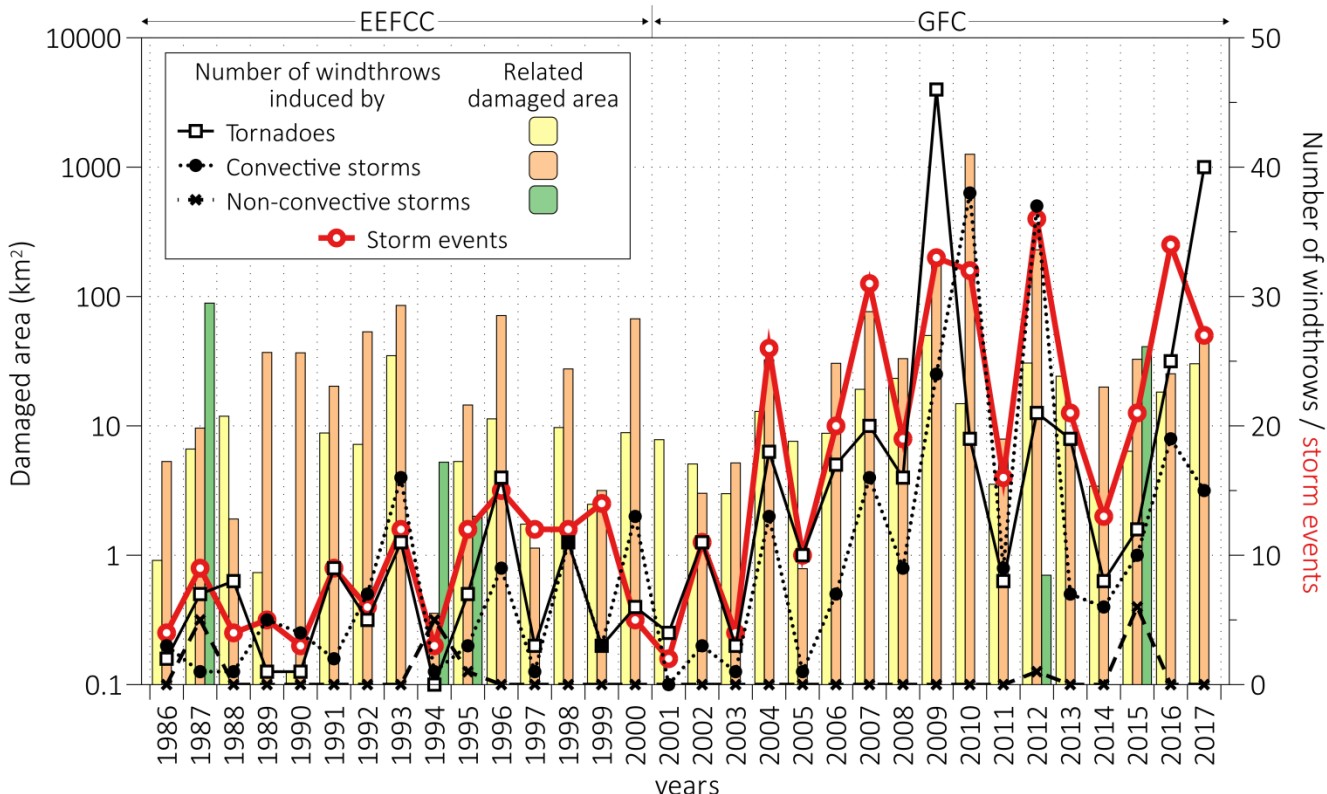

**Figure 11: Interannual variability of the number of windthrows, related damaged area, and number of storm events.** Note the logarithmic scale for the damaged area. Periods for the EEFCC and GFC datasets are indicated.


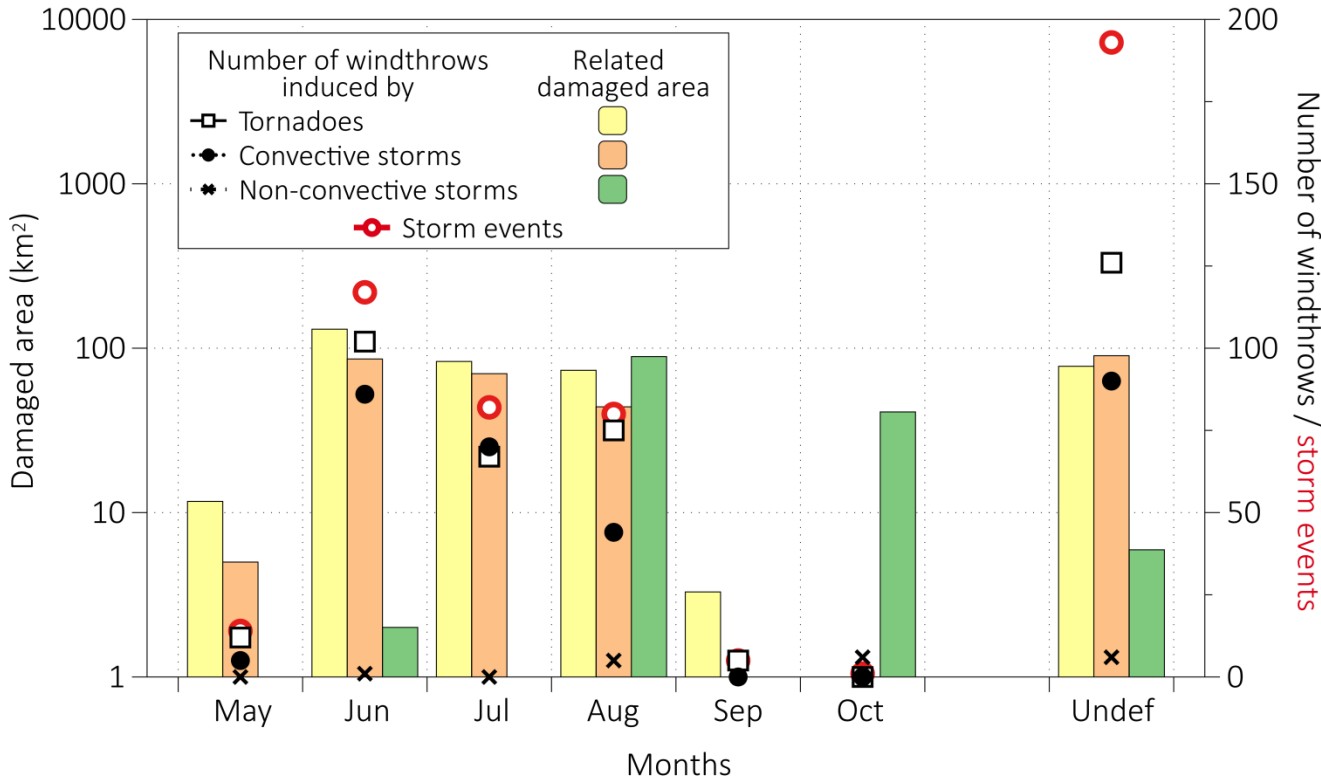

**Figure 12: Annual cycle of the number of windthrows, related damaged area, and number of storm events.** Note the logarithmic scale for damaged area.



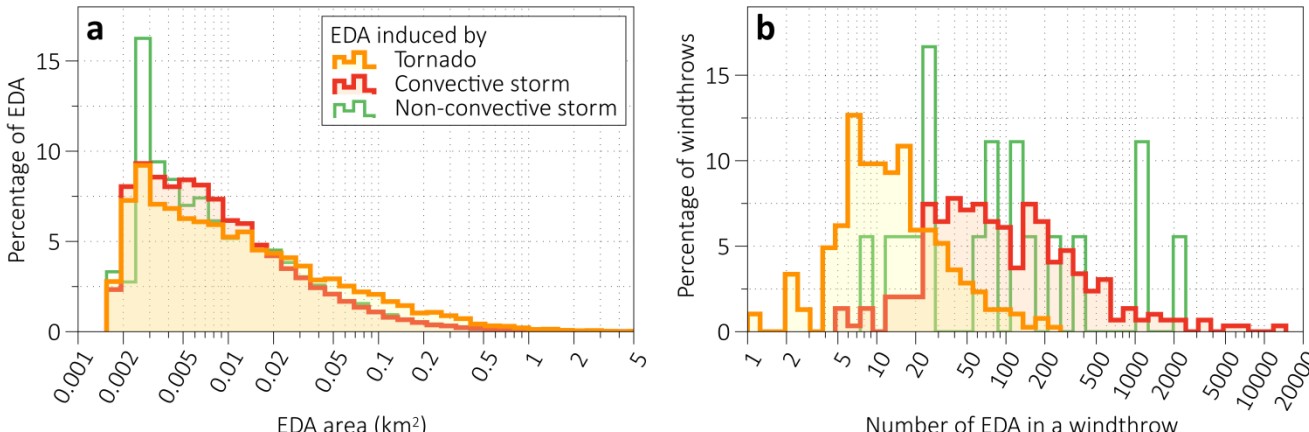


**Figure 13: Distribution of (a) size of EDAs for different types of windthrows and of (b) a number of EDAs within one windthrow.**





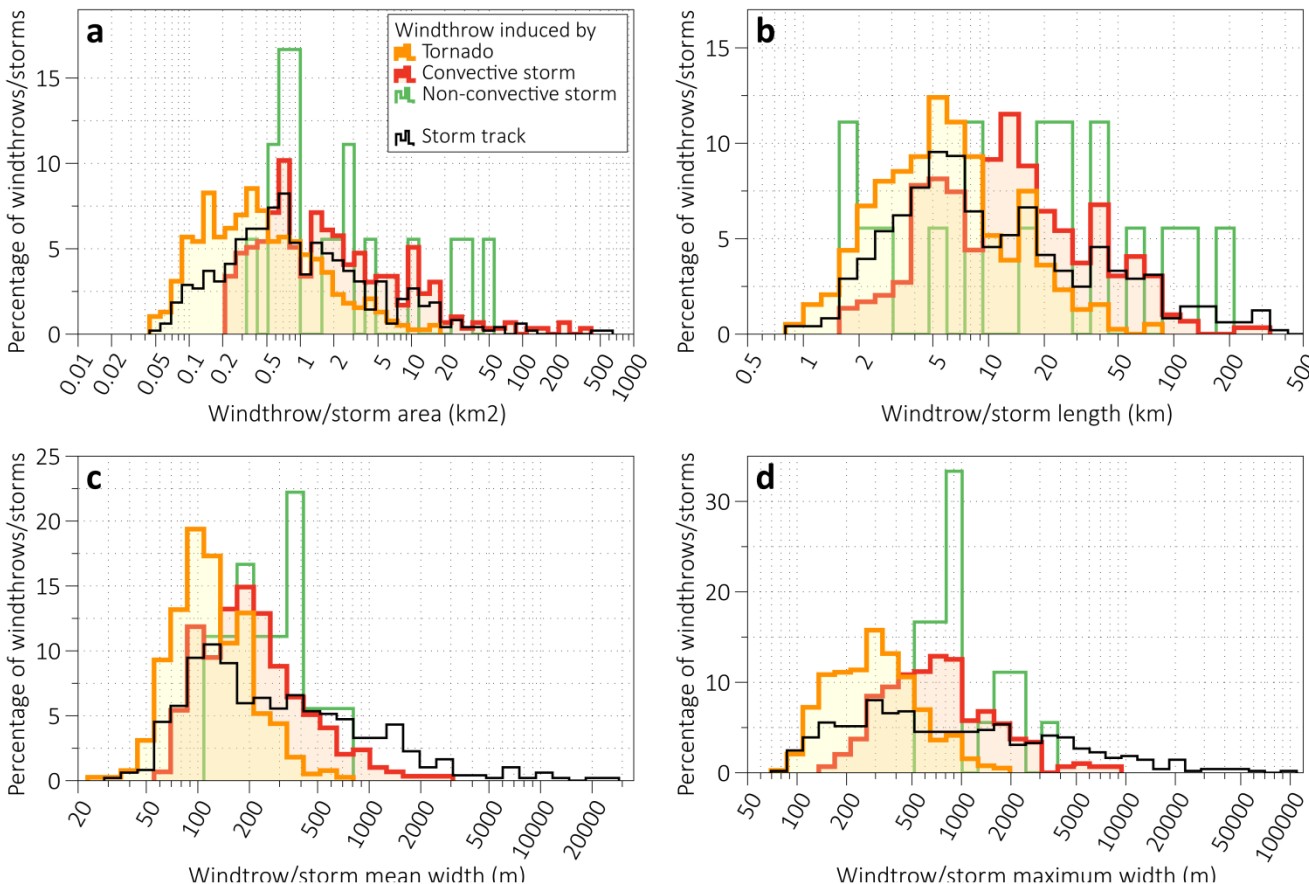

**Figure 14: Distribution of geometric parameters of windthrows of different types and storm tracks:** (a) area, (b) length, (c) mean width, and (d) maximum width.