# Peer review of "A satellite-derived database for stand-replacing windthrow events in boreal forests of European Russia in 1986–2017"

_Earth System Science Data, 2020_

## Referee Comment (RC1) · Anonymous Referee #1 · 22 Jun 2020

The manuscript "A satellite-derived database for stand-replacing windthrows in boreal forests of the European Russia in 1986–2017" present a GIS-database on storm events in European Russia. The data base spans more than 30 years and contains over 100,000 entries, an enormous amount of data! As such, I believe that the manuscript and database are an important addition to the literature, as information on windthrows is rare. I applaud the authors for undertaking such a great effort in compiling the data. That said, I have some issues and comments, which in my opinion need to be addressed before publication.

(1) The assessment includes a lot of subjective calls from the interpreter and there is

no formal validation. I fully understand that validating the database is very challenging, but I would have at least expected some assessment on how accurate the data is. There are many steps involved in collecting the data, and some of them seem to be highly dependent on local knowledge. From this I assume that the accuracy will be higher in some regions (where there is ample local knowledge), but lower in others. In particular, I was wondering whether you searched the forest area of ER systematically or whether you applied any other "sampling strategy" to ensure you don't miss a storm event. Moreover, the accuracy will depend on the availability of HRI. Was HRI available for each year after 2001, or only for selected years? The authors address some of these issues in the discussion, which I appreciated, but maybe they should make clear from the beginning on that the database is a subjective collection and probably far from complete and/or consistent.

(2) In the same line as the previous comment, many of the decision to group patches to windthrows, etc. are based on arbitrary thresholds. I wonder whether the authors have tested the sensitivity of their results to those thresholds. The allocation of patches to windthrow events might look very different with slight changes in threshold values.

(3) You do not explain how the Landsat data was processes. Please give some details on the processing. Moreover, how many Landsat images were available per year, on average? I guess that in Russia many winter images are covered by snow. As you use pre/post windthrow Landsat images to ensure windthrow detection, data availability is crucial. In years with only few observations, detection accuracy might be lower.

(4) The manuscript is already well written, but needs some language editing. I give some suggestions below. Also, it is quite dry to read in some parts, but I guess this is typical for a data paper.

Specific comments:

Throughout the manuscript: No "the" needed before "European Russia".

L. 17: Sentence starting with "Additional...": Something wrong with the sentence, please revise (e.g., "Additional information, such as ..., is also provided."

L. 21: Change to "..., which is in contrast to ...".

L. 23: Change to "... can be used by both science and management."

L. 28: "Forests are..." L. 29: "Exposed to..." and replace "and windstorms" to "or windstorms".

L. 37: "..., and droughts..."

L. 38: Remove "the" before "Western..."

L. 41: Change to "...like increasing growing stock..."

L. 43: Remove "as well".

L. 46: Remove "of" before "wind-related ..."

L. 56: Remove "substantially"

L. 62: Replace "on the Earth" with "globally"

L. 66: Replace "publication" with "opening"

L. 75: Replace "the archive of Landsat images" with "the Landsat archive"

L. 76: Remove "the" before "public map..."

L. 79: Add "the" before "windthrow delineation..."

L. 74-82: You rather describe where you do what in the manuscript, which is not of interest at this point. I strongly suggest to revise this paragraph to give a detailed description of what data and which specific attributes you will collect; and what is the rational for collecting those.

L. 88: "study region" not "study regions". It is also not clear to me what the following

sentence means. Please revise.

L. 93: What is "the large area"? Do you mean a specific area?

L. 100ff: Why is this written in bullet points? I strongly suggest to write the section out as proper text.

L. 101/102: Change to ". . . forest disturbance at annual temporal resolution."

L. 108: Change to ". . . on forest loss classified into. . ."

L. 122: How do you download images from Google Maps or Bing Maps?

L. 175: This might be a personal flavor, but could you give the areas in hectare or square meters? A value of 0.0018 km2 is hard to image, given all the leading zeros.

Section 4.1.1.: How did you ensure you didn't miss a windthrow? Was there some systematic sampling design applied? How can you be sure there are only 450 windthrows? Did you look any each and every disturbance patch?

L. 195: Revise beginning of the sentence.

L. 200: How was the Landsat data processed? Were any corrections or masking algorithms applied? More information needed.

L. 211: Change "the stand-replacing" to "a stand-replacing"

L. 212: Not sure what is meant by "However, this value may be less if these disturbances hold the substantial part of the image."? Please explain.

L. 230: "A similar threshold value. . ."

L. 230ff: Here for instance I have the feeling that the choice of thresholds is very subjective and depends largely on the interpreter and his/her knowledge. How can we make sure that the data collection is systematic and not biased?

Note: I stopped marking every language issue. Please do a proper language check

before resubmitting.

L. 270: How did you assess whether a storm happened in winter? My guess is that satellite data and HRI data is mostly only available for the growing period.

L. 292ff: How was the central line drawn? By hand? Or did you use an algorithm to do so? Is the line sensitive to the allocation of patches to wind throw events?

L. 389ff: This is discussion and not results. In genera you mix up a lot of results reporting and discussion. In general, I think that is fine for such a paper, but then you should rename the section to Results and Discussion; and give the Discussion section a more informed title.

L. 446: You note that no winter windthrows were found. This might be a result of missing winter observations, right?

Conclusion section: You are cautious in interpreting trends in windthrow area due to inconsistencies in your data. This is great and much appreciated. However, you then conclude that the positive trend for large windthrows likely is "real". But how do you come to this conclusion? Without proper validation, we can't be sure about this. I suggest to remove this conclusion.

Comments on the database:

The dataset was downloadable and could be opened in standard GIS software (QGIS). All entries had attributes. The projection of the GIS layers was set. Units were not specific. I thus encourage the authors to add a README or similar metadata file to describe the units (i.e., are, length, etc.).

---

## Short Comment (SC1) · 4 Jul 2020

This is a really impressive paper that substantially adds to our knowledge of the importance of wind as a disturbance agent in boreal forests. It also makes a beautiful complementary study to the recent publication of Forzieri et al (2020) "A spatially explicit database of wind disturbances in European forests over the period 2000 – 2018" also in Earth System Science Data 10.5194/essd-12-257-2020 The paper needs to be published. However, there are some points that need to be addressed that I will comment on separately.

[Figure]

2020.

---

## Short Comment (SC2) · 5 Jul 2020

Main Points

1. Language needs some help. I would recommend the paper being checked by a native English paper because in some places the meaning was not totally clear. 2. Unlike the earlier reviewer I was really impressed at how much effort was put into assessing the potential errors in the assessment. I also think the authors were completely honest about limitations in the data and their assessments. Such honesty will give really help users of the data.

Minor Points 1. I would recommend referencing the paper Ulanova, N.G., 2000. The effects of windthrow on forests at different spatial scales: a review. For. Ecol. Manage. 135, 155–167. https://doi.org/10.1016/S0378-1127(00)00307-8 This paper deals with the effects of wind in the Russian boreal forest. It is a brilliant and seminal paper. 2. "windthrow" not "windthrows" throughout. Plurals in English can be difficult. 3. I think early on you need to tell the reader that your analysis was done manually and not automated. That is important to understand the process. Also refer whenever possible to Fig. 2 which is super useful. 4. Page 6. I think some more justification of the limits you set would be helpful, for example the lower limits on size of EDAs. 5. Page 7. What did you do in highly populated areas? Did you just accept that you will have missed wind damage on some occasions? 6. Page 7, line 212. Use "represent" not "hold". 7. Page 8, line 222. Again justify the figure of 5-10 km. Do you use "rules" or is this judgement left to the observer? 8. Page 9, line 261. I think "predominately" is better than "prevalently". 9. Page 11, line 346. I think the symbol should be $\sim$ rather than $<$ 10. Page 13, line 367. When you say "less than 13%" what are you referring to? 13% of what? 11. Page 13, line 392. But trees could break. You don't manage wind break at all in your paper. 12. Page 13, line 398. Please define "parallel" and "successive". 13. Page 14, line 417. What do you mean by dark-coniferous? What species are you referring to? 14. Page 14, line 430. Too coarse for what? Correlation? With what? 15. Page 15, line 461. Write out local time not LT. 16. Page 490. Reference work in USA and Amazon on damage from derechos or squall lines. See a) Negrón-Juárez, R.I., Chambers, J.Q., Guimaraes, G., Zeng, H., Raupp, C.F.M., Marra, D.M., Ribeiro, G.H.P.M., Saatchi, S.S., Nelson, B.W., Higuchi, N., 2010. Widespread Amazon forest tree mortality from a single cross-basin squall line event. Geophys. Res. Lett. 37, 1–5. https://doi.org/10.1029/2010GL043733 and b) Peterson, C. J., 2000: Catastrophic wind damage to North American forests and the potential impact of climate change. Sci. Total Environ.,262, 287–311. 17.

---

## Referee Comment (RC2) · Anonymous Referee #2 · 20 Jul 2020

The manuscript described the development of a satellite-driven GIS database containing information on stand-replacing windthrows in boreal forests of the European Russia between 1986 and 2017. Based on this database the spatial and temporal distribution of windthrows in European Russia is also presented in the manuscript. The manuscript is well structured and written (although the large number of acronyms make the reading difficult in places) and is free of errors in logic. Furthermore, the manuscript is filling a gap concerning the climatology of severe storms in European Russia, especially in the sparsely populated area.

Minor comments

line 10: "natural disturbances" is not clearly defined in this context

line 14: "[...] determine the type of"

line 17: "[...] with an area"

line 18: replace "contained" with "containing"

line 21: "[...] happened in the summer"

line 24: no need to include this information in the abstract.

line 29: from my point of view "disturbance" is not a good choice of word in this context

line 36: "Particularly, both the [...]"

line 45: "[...] convective storms in the warm season"

line 50: not sure what "macro-regional" is in this context

line 52: replace "presented" with "compiled"

line 54: remove extra open parenthesis

line 100: "[...] that occurred"

lines 151-155: maybe the authors should add a figure containing examples of the "three hierarchical levels"

line 208-210: the choice of thresholds is not clearly defined

lines 220-240: is not clear how the thresholds used in this section have been derived

line 261: the ratio of length to width of 10:1 is based on previous studies conducted for European Russia?

line 323: "[...] we calculated as well"

line 323: "estimation" instead of "estimating"

[Figure]

line 348: citation not in the correct form

line 361: citation not in the correct form

line 439: "storm events occurred"

line 472: is not clear what the authors meant by "contain fewer plots"

———————————————

---

## Author Comment (AC1) · 26 Aug 2020

We would like to thank two anonymous reviewers and Dr. Barry Gardiner for their valuable and constructive comments on our manuscript. Following the suggestions of reviewers, we made a revision of the manuscript including clarifying some aspects of the data collection process, correction of grammar and syntax errors, implementation of minor revisions. In particular, we have renamed sections, added a new figure. We have uploaded the new version of the dataset with changed objects numbering and with the readme file that describes the structure of attribute tables and the units. Because of language correction, we have changed the title of the manuscript to 'A satellite-

derived database for stand-replacing windthrow events in boreal forests of European Russia in 1986–2017'. The point–to–point answers on reviewers' comments are listed in supplement .pdf file

Anonymous Referee 1: The manuscript "A satellite-derived database for stand-replacing windthrows in boreal forests of the European Russia in 1986–2017" present a GIS-database on storm events in European Russia. The data base spans more than 30 years and contains over 100,000 entries, an enormous amount of data! As such, I believe that the manuscript and database are an important addition to the literature, as information on windthrows is rare. I applaud the authors for undertaking such a great effort in compiling the data. That said, I have some issues and comments, which in my opinion need to be addressed before publication. (1) The assessment includes a lot of subjective calls from the interpreter and there is no formal validation. I fully understand that validating the database is very challenging, but I would have at least expected some assessment on how accurate the data is. There are many steps involved in collecting the data, and some of them seem to be highly dependent on local knowledge. From this I assume that the accuracy will be higher in some regions (where there is ample local knowledge), but lower in others. In particular, I was wondering whether you searched the forest area of ER systematically or whether you applied any other "sampling strategy" to ensure you don't miss a storm event. Moreover, the accuracy will depend on the availability of HRI. Was HRI available for each year after 2001, or only for selected years? The authors address some of these issues in the discussion, which I appreciated, but maybe they should make clear from the beginning on that the database is a subjective collection and probably far from complete and/or consistent. We agree that due to several limitations of the method and satellite data the database is spatially and temporally inhomogeneous and hence incomplete. We have added corresponding information into the Abstract. Sections 2, 4, and 6 were also amended. We devoted the whole Section 6 to discussing of factors influencing the data accuracy including such factors as percentage of forest-covered area, forests species composition and forest management practices. In general, our data has highest accuracy for

low-populated northern and eastern part of the ER, where forests cover 70-90Each windthrow from our dataset was validated (that is, it was clarified that it is actually a windthrow), based on pre- and post-event Landsat/Sentinel-2 images, high-resolution images and additional information. So, the probability that any forest disturbance was mistakenly referred to a windthrow is minimal. However, we agree with the reviewer, that some thresholds used in the data collection process, in particular the thresholds for the minimum area of EDAs and windthrow, as well as the threshold for the minimum distance for separating successive windthrow, are somewhat subjective. We should stress, that the searching of windthrow areas based on GFC/EEFCC was systematic. The GFC/EEFCC-based collection of forest loss areas with windthrow-like signatures was carried out separately for each region of the ER. A grid with 50 km cell size was built inside the region, which helped to organize the searching of windthrow-like forest disturbances. The relevant information has been added to Section 4.1.1. The HRI are available only for several years (usually 2-8 images for the entire period 2001-2017), and the year of 2001 is the only year of the appearance for the first HRI. Wherein, some areas in the northern part of the ER are not covered by HRI. However, the lack of HRI affects determination of windthrow type (windstorm- or tornado-induced) rather its identification accuracy. The relevant information has been added to Section 2.2.

(2) In the same line as the previous comment, many of the decision to group patches to windthrows, etc. are based on arbitrary thresholds. I wonder whether the authors have tested the sensitivity of their results to those thresholds. The allocation of patches to windthrow events might look very different with slight changes in threshold values. Indeed, in our method, we used a number of thresholds that have some subjectivity but are based on previous studies. For instance, the 10-km threshold for separate successive windthrow from each other is based on study of Doswell and Burgess (1988), who proposed the 5–10 miles (8–16 km) threshold for the gap to separate one skipping tornado from two successive tornadoes. The threshold for minimum area of EDAs was chosen based on study of Koroleva and Ershov (2012) who showed high uncertainty of estimated geometrical characteristics of small-scale windthrow (less than 1800 m2,

i.e., two GFC pixels). In addition, we decided to filter out such small-scale disturbances since it is virtually impossible to confirm their wind-related origin. We performed sensitivity test for the latter threshold and found that the absence of minimum accepted area for EDAs will increase area of windthrow by 2-3We have added this information to Sections 4.1.1 and 4.1.4.

(3) You do not explain how the Landsat data was processes. Please give some details on the processing. We downloaded Landsat images (L1T processing level) from https://earthexplorer.usgs.gov/ and https://eos.com/landviewer. We did not use any atmospheric correction algorithm for the image preprocessing (see our rationale below). For NDII-based delineation process, we used only images with cloudiness less than 10The relevant description been added to the revised version of the manuscript (to different parts of Section 4). The reference 'Foga et al., 2017' has been added to the reference list. Moreover, how many Landsat images were available per year, on average? The availability of cloudless Landsat images varied from year to year. The lowest number of cloud-free images (2-4 images a year on average) is available for 2003-2006 and 2012, when only Landsat-7 (SLC-off) data were available (Potapov et al., 2015). Hence, the worst accuracy of windthrow date determination is typical for these years. On average, 8-10 cloud-free images per year can be used for windthrow identification and dates determination. Due to Sentinel-2A satellite launching, number of images per year had an abrupt increase after the summer of 2016. We have added this information to Section 4.4. I guess that in Russia many winter images are covered by snow. As you use pre/post windthrow Landsat images to ensure windthrow detection, data availability is crucial. In years with only few observations, detection accuracy might be lower. Winter images (of land covered with snow) were successfully used for windthrow identification, especially if a storm occurred at the end of summer season, and autumn season lacked cloud-free images. (4) The manuscript is already well written, but needs some language editing. I give some suggestions below. Also, it is quite dry to read in some parts, but I guess this is typical for a data paper. We thank the reviewer for his suggestions on language editing.

Specific comments: Throughout the manuscript: No "the" needed before "European Russia". Corrected (including the title). L. 17: Sentence starting with "Additional. . .": Something wrong with the sentence, please revise (e.g., "Additional information, such as . . ., is also provided." Corrected. L. 21: Change to ". . ., which is in contrast to . . .". Corrected. L. 23: Change to ". . . can be used by both science and management." Corrected. L. 28: "Forests are. . ." Corrected. L. 29: "Exposed to. . ." and replace "and windstorms" to "or windstorms". Corrected. L. 37: ". . ., and droughts. . ." Corrected. L. 38: Remove "the" before "Western. . ." Corrected. L. 41: Change to ". . .like increasing growing stock. . ." Corrected. L. 43: Remove "as well". Corrected. L. 46: Remove "of" before "wind-related . . ." Corrected. L. 56: Remove "substantially" Corrected. L. 62: Replace "on the Earth" with "globally" Corrected. L. 66: Replace "publication" with "opening" Corrected. L. 75: Replace "the archive of Landsat images" with "the Landsat archive" Corrected. L. 76: Remove "the" before "public map. . ." Corrected. L. 79: Add "the" before "windthrow delineation. . ." Corrected. L. 74-82: You rather describe where you do what in the manuscript, which is not of interest at this point. I strongly suggest to revise this paragraph to give a detailed description of what data and which specific attributes you will collect; and what is the rational for collecting those. Corrected. The paragraph has been partially rewritten; the description of the data and specific attributes has been added. The sentence on the importance of collecting the windthrow database has been added to the previous paragraph. L. 88: "study region" not "study regions". It is also not clear to me what the following sentence means. Please revise. Corrected, the second part of the sentence has been removed. L. 93: What is "the large area"? Do you mean a specific area? Corrected. L. 100ff: Why is this written in bullet points? I strongly suggest to write the section out as proper text. Corrected. L. 101/102: Change to ". . . forest disturbance at annual temporal resolution." Corrected. L. 108: Change to ". . . on forest loss classified into. . ." Corrected. L. 122: How do you download images from Google Maps or Bing Maps? The sentence has been completely rewritten. L. 175: This might be a personal flavor, but could you give the areas in hectare or square meters? A value of 0.0018

km2 is hard to image, given all the leading zeros. Corrected. Section 4.1.1.: How did you ensure you didn't miss a windthrow? Was there some systematic sampling design applied? How can you be sure there are only 450 windthrows? Did you look any each and every disturbance patch? We assume that the searching of windthrow areas based on GFC/EEFCC was rather systematic. The GFC/EEFCC-based collection of forest loss areas with windthrow-like signatures was carried out separately for each region of the Russian Federation. A grid with 50 km cell size was built inside each region, which helped to organize the searching of windthrow-like forest disturbances — the searching was performed sequentially in each cell of the grid. However, we agree, that some windthrow areas can be missed and discuss it in Section 6. Moreover, since we used the threshold values of windthrow area (0.05 km2 for tornado-induced windthrow areas and 0.25 km2 for other windthrow), all forest disturbances with smaller area are missed in our database. The relevant information has been added to Section 4.1.1.

L. 195: Revise beginning of the sentence. Corrected. L. 200: How was the Landsat data processed? Were any corrections or masking algorithms applied? More information needed. We applied none atmospheric correction algorithm for preprocessing Landsat images, since NDII is based on the near-infrared (0.76 - 0.90 nm) and middle-infrared (1.55 - 1.75 nm) spectral bands that are almost insensitive to atmospheric impact. We used images with cloudiness less than 10L. 211: Change "the stand-replacing" to "a stand-replacing" Corrected. L. 212: Not sure what is meant by "However, this value may be less if these disturbances hold the substantial part of the image."? Please explain. The paragraph has been rewritten for clarity. We estimated threshold value from the statistics of $\Delta$NDII raster. Firstly, we obtained the mean value and standard deviation of $\Delta$NDII within the entire forest-covered area on image. Stand-replacing forest disturbance inherently has $\Delta$NDII values substantially higher than the image average. To separate stand-replacing forest disturbance from other forest-covered area, we used the threshold value of two standard deviations, which was previously tested by Koroleva and Ershov (2012). However, in some cases

the ΔNDII distribution within the entire image was skewed (e.g., due to the presence of cloud decks or haze on the post-event image). In such cases, we lowered the threshold value of ΔNDII iteratively by comparing the detected changes with results of visual identification of windthrow on a post-event image (using several examples located in different parts of windthrow). As a result, actual threshold values ranged from 1.5 to 2 standard deviations. Then, a binary raster of detected changes (i.e., forest losses) has been created (see fig. 4d) and converted to a shapefile. This information has been added to Section 4.1.3.

L. 230: "A similar threshold value. . ." Corrected. L. 230ff: Here for instance I have the feeling that the choice of thresholds is very subjective and depends largely on the interpreter and his/her knowledge. How can we make sure that the data collection is systematic and not biased? We selected this threshold values based on Doswell and Burgess (1988), who proposed the 5-10 miles (8-16 km threshold for the gap length to separate one skipping tornado from two successive tornadoes. It is also important, that this threshold determines only geometrical characteristics of single windthrow (the second layer of the GIS database) and do not affect the total area of a storm event (the third layer of the GIS database). The relevant text has been added to Sections 4.1.4 and 4.3. L. 270: How did you assess whether a storm happened in winter? My guess is that satellite data and HRI data is mostly only available for the growing period. The availability of the Landsat and Sentinel-2 images does not depend on the season of a year, excluding some years, e.g. 2003–2006, when only the Landsat-7 (SLC-off) images were available. In fact, wintertime images were widely used at all stages of the data collection. We agree with the reviewer that the frequency of obtaining of cloudless images in autumn and winter was lower than in summer season, but it was sufficient for the analysis. The text in Section 4.4 has been corrected to highlight this issue. L. 292ff: How was the central line drawn? By hand? Or did you use an algorithm to do so? Is the line sensitive to the allocation of patches to wind throw events? The central line was created automatically (using a Python tool) as a distance between two farthest points of a windthrow. It is insensitive to the allocation of patches to windthrow

area. The explanation has been added to Section 4.3. L. 389ff: This is discussion and not results. In genera you mix up a lot of results reporting and discussion. In general, I think that is fine for such a paper, but then you should rename the section to Results and Discussion; and give the Discussion section a more informed title. We agree. The section "Results" has been renamed to "Results and Discussion", and the section "Discussion" has been renamed to "Data and method limitations" L. 446: You note that no winter windthrows were found. This might be a result of missing winter observations, right? Both Landsat-based products GFC and EEFCC reveals stand-replacing windthrow area regardless of the season of its appearance. In particular, if windthrow happened in winter it would be clearly seen on image taken in subsequent vegetation period because of rather slow forest recovery process. Therefore, the revealed lack of winter windthrow is feasible due to the climatic conditions of the study area and does not associated with data limitations. In particular, winter storms from Western Europe reach the territory of Russia already weakened (Haylock, 2011), while low temperatures and soil freezing also prevent stand-replacing windthrow in Russian forests during winter season (Suvanto et al., 2016). According to (Suvanto et al., 2016), winter windthrow are not typical for Finland as well. The relevant information has been added to Section 5.3. Conclusion section: You are cautious in interpreting trends in windthrow area due to inconsistencies in your data. This is great and much appreciated. However, you then conclude that the positive trend for large windthrows likely is "real". But how do you come to this conclusion? Without proper validation, we can't be sure about this. I suggest to remove this conclusion. We agree with this suggestion and have removed this conclusion. Comments on the database: The dataset was downloadable and could be opened in standard GIS software (QGIS).All entries had attributes. The projection of the GIS layers was set. Units were not specific. I thus encourage the authors to add a README or similar metadata file to describe the units (i.e., are, length, etc.). We agree. The 'Readme' file that describes the structure of attribute tables and the units has been added to the dataset. We have also changed the objects numbering inside each GIS layer of the database (the relevant description

has been added to Section 3). The URL address of the dataset has been changed to https://doi.org/10.6084/m9.figshare.12073278.v6

  Anonymous Referee 2: The manuscript described the development of a satellite-driven GIS database containing information on stand-replacing windthrows in boreal forests of the European Russia between 1986 and 2017. Based on this database the spatial and temporal distribution of windthrows in European Russia is also presented in the manuscript. The manuscript is well structured and written (although the large number of acronyms make the reading difficult in places) and is free of errors in logic. Furthermore, the manuscript is filling a gap concerning the climatology of severe storms in European Russia, especially in the sparsely populated area. Minor comments line 10: "natural disturbances" is not clearly defined in this context The word "natural" has been deleted. line 14: "[...] determine the type of" Corrected. line 17: "[...] with an area" Corrected. line 18: replace "contained" with "containing" Corrected according to the suggestion of the first reviewer. The phrase has been modified as follows: "Additional information such as weather station reports and event description from media sources is also provided". line 21: "[...] happened in the summer" Corrected. line 24: no need to include this information in the abstract. We agree; the information has been removed. line 29: from my point of view "disturbance" is not a good choice of word in this context 'Natural disturbance agent' is a commonly used term relating to wildfires, windstorms, insect outbreaks, etc. (see e.g. Ulanova, 2000; Seidl et al., 2017). line 36: "Particularly, both the [...]" Corrected. line 45: "[...] convective storms in the warm season" Corrected. line 50: not sure what "macro-regional" is in this context The word 'macro-regional' is has been replaced by 'international'. line 52: replace "presented" with "compiled" Corrected. line 54: remove extra open parenthesis Corrected. line 100: "[...] that occurred" Corrected. lines 151-155: maybe the authors should add a figure containing examples of the "three hierarchical levels" A new figure 3 has been added that shows an example with all three hierarchical levels of the database, explains the determination of geometric characteristics of storm event, and shows the examples of parallel and successive windthrow. Correspondingly, numbers of all subsequent figures have been changed. line 208-210: the choice of thresholds is not clearly defined The paragraph has been rewritten for clarity. We estimated threshold value from the statistics of ∆NDII raster. Firstly, we obtained the mean value and standard deviation of ∆NDII within the entire forest-covered area on image. Stand-replacing forest disturbance inherently has ∆NDII values substantially higher than the image average. To separate stand-replacing forest disturbance from other forest-covered area, we used the threshold value of two standard deviations, which was previously tested by Koroleva and Ershov (2012). However, in some cases the ∆NDII distribution within the entire image was skewed (e.g., due to the presence of cloud decks or haze on the post-event image). In such cases, we lowered the threshold value of ∆NDII iteratively by comparing the detected changes with results of visual identification of windthrow on a post-event image (using several examples located in different parts of windthrow). As a result, actual threshold values ranged from 1.5 to 2 standard deviations. Then, a binary raster of detected changes (i.e., forest losses) has been created (see fig. 4d) and converted to a shapefile. This information has been added to Section 4.1.3.

lines 220-240: is not clear how the thresholds used in this section have been derived We used the 10-km threshold as the slightly modified value proposed by Doswell and Burgess (1988) who suggested to use the 5–10 miles (8–16 km) threshold for a gap to discriminate between one skipping tornado and two successive tornadoes. The reference to (Doswell and Burgess, 1988) has been added to the revised version of the manuscript. line 261: the ratio of length to width of 10:1 is based on previous studies conducted for European Russia? Yes. This is typical ratio for tornado length and width for US (Schaefer and Edwards, 1999) and for Northern Eurasia (Shikhov and Chernokulsky, 2018). The clarification has been added to the revised version of the manuscript. line 323: "[...] we calculated as well" Corrected. line 323: "estimation" instead of "estimating" Corrected. line 348: citation not in the correct form We have checked the correctness of the citation. line 361: citation not in the correct form We have checked the correctness of the citation. line 439: "storm events occurred" Corrected. line 472: is not clear what the authors meant by "contain fewer plots" A typical tornado-induced windthrow event on average contains less EDAs than non-tornado induced one. The word "Plots" has been replaced by "EDAs".   Interactive comment from Barry Gardiner Main Points 1. Language needs some help. I would recommend the paper being checked by a native English paper because in some places the meaning was not totally clear. Language has been improved. 2. Unlike the earlier reviewer I was really impressed at how much effort was put into assessing the potential errors in the assessment. I also think the authors were completely honest about limitations in the data and their assessments. Such honesty will give really help users of the data. In the revised version, we made minor changes according to the first reviewer suggestions. Minor Points 1. I would recommend referencing the paper Ulanova, N.G., 2000. The effects of windthrow on forests at different spatial scales: a review. For. Ecol. Manage.135, 155–167. https://doi.org/10.1016/S0378-1127(00)00307-8 This paper deals with the effects of wind in the Russian boreal forest. It is a brilliant and seminal paper. The reference to this paper has been added in the Introduction section. 2."windthrow" not "windthrows" throughout. Plurals in English can be difficult. Corrected (including the title). 3. I think early on you need to tell the reader that your analysis was done manually and not automated. That is important to understand the process. Also refer whenever possible to Fig. 2 which is super useful. We have added the word 'manually' into the Abstract. Additionally, the short sentence on the manual character of the data analysis has been added to the beginning of Section 4. The additional references to figure 2 have been added throughout the text. 4. Page 6. I think some more justification of the limits you set would be helpful, for example the lower limits on size of EDAs. We removed all EDAs with an area $\leq$ 1800 m2 that equals to area of two GFC pixels. We filtered out such small-scale disturbances since it is virtually impossible to confirm their wind-related origin. Moreover, the area of local windthrow can be almost three times overestimated by Landsat images (Koroleva and Ershov, 2012). Thus, this is the balance between slight underestimation of the total area of windthrow (by 2-3We have added this rationale to Section 4.1.1. 5. Page 7. What did you do in highly populated areas? Did you just accept that you will have missed wind damage on some occasions? Indeed, the windthrow data obtained for the period before 2000 (using the EEFCC dataset) may be incomplete for highly-populated regions of the ER due to assignment of forest losses to broad periods, i.e., 1986–1988 and 1989–2000. To partially avoid missing of windthrows, using Landsat images, we performed additional verification of all large-scale forest loss areas (with area more than 5 km2) in these regions independently of their geometry, since windthrow areas can be totally masked out by logged areas. Thus, we were able to find three large-scale windthrow events in highly-populated regions of the ER. However, some windthrow events can still be missed. The clarification has been added to Section 4.1.2. 6. Page 7, line 212. Use "represent" not "hold". The paragraph has been completely rewritten according to the suggestions of the first reviewer. 7. Page 8, line 222. Again justify the figure of 5-10 km. Do you use "rules" or is this judgement left to the observer? We used the 10 km threshold, which is in the range of 8–16 km (5–10 miles) proposed by Doswell and Burgess (1988) to discriminate between one skipping tornado and two successive tornadoes. Therefore, if the nearest EDAs were located at a distance more than 10 km from each other, they belonged to different windthrow. If the distance between them was less or equal to 10 km, they belong to one windthrow except for several cases. The exceptions were associated with changes of windthrow direction, transformations of one windthrow type to another identified by the HRI, and abrupt change of forest damage degree. 8. Page 9, line 261. I think "predominately" is better than "prevalently". Corrected. 9. Page 11, line 346. I think the symbol should be âĹij rather than < Corrected. 10. Page 13, line 367. When you say "less than 13It is less than 1311. Page 13, line 392. But trees could break. You don't manage wind break at all in your paper. It is of note, that we cannot determine whether the trees were felled or broken by the wind based on satellite images, even having very high resolution. Therefore, we use a single term "windthrow" for all types of wind-induced forest damage. This clarification has been added to the beginning of Section 5. 12. Page 13, line 398. Please define "parallel" and "successive". Successive windthrow areas induced by one storm event

follow downwind one after another and approximately fall on one straight line (the angle of deviation from this line does not exceed 10-20°). Such windthrow are presumably induced by one convective cell generating a sequence of squalls or tornadoes. In contrast, parallel windthrow areas that located within one storm event are situated parallel to each other (with an angle less than 30°). They are presumably associated with two or more different convective cells or mesocyclones, generating squalls or tornadoes, often embedded into one mesoscale convective system. The examples of parallel and successive windthrow are shown at new Fig. 3. The definitions have been added to Section 4.1.4. 13. Page 14, line 417. What do you mean by dark-coniferous? What species are you referring to? Dark-coniferous forests in the ER consist of three dominating tree species such as Picea abies, Picea obovata and Ábies sibírica. We have added this information to section 2.1. 14. Page 14, line 430. Too coarse for what? Correlation? With what? The sentence has been deleted. Instead, we highlighted, that using out dataset, estimates of relationship between windthrow area and forest stands characteristics can be carried out in future studies at a regional scale. 15. Page 15, line 461. Write out local time not LT. Corrected. 16. Page 490. Reference work in USA and Amazon on damage from derechos or squall lines. See a) Negrón-Juárez, R.I., Chambers, J.Q., Guimaraes, G., Zeng, H., Raupp, C.F.M., Marra, D.M., Ribeiro, G.H.P.M., Saatchi, S.S., Nelson, B.W., Higuchi, N., 2010. Widespread Amazon forest tree mortality from a single cross-basin squall line event. Geophys. Res. Lett. 37, 1–5. https://doi.org/10.1029/2010GL043733 and b) Peterson, C. J., 2000: Catastrophic wind damage to North American forests and the potential impact of climate change. Sci. Total Environ.,262, 287–311. Both references have been added.

References: Doswell, C.A. and Burgess, D.W.: On some issues of United States tornado climatology. Monthly Weather Review, 116, 495–501, 1988. Foga, S., Scaramuzza, P.L., Guo, S., Zhu, Z., Dilley, R.D., Beckmann, T., Schmidt, G.L., Dwyer, J.L., Hughes, M.J., and Laue, B. Cloud detection algorithm comparison and validation for operational Landsat data products. Remote Sensing of Environment, 194, 379-390. http://doi.org/10.1016/j.rse.2017.03.026, 2017 Haylock, M,
R.: European extra-tropical storm damage risk from a multi-model ensemble of dynamically-downscaled global climate models. Natural Hazards and Earth System Sciences, 11, 2847–2857, doi:10.5194/nhess-11-2847-2011, 2011. Koroleva, N. V., and Ershov, D. V.: Estimation of error in determining the forest windfall disturbances area on high spatial resolution space images of LANDSAT-TM. In: Current Problems in Remote Sensing of the Earth From Space, 9, 80–86, 2012. (in Russian) Schaefer, J.T., and Edwards, R., The SPC tornado/severe thunderstorm database. In: Preprints, 11th Conf. on Applied Climatology. Amer. Meteor. Soc, Dallas, TX Available online at. https://ams.confex.com/ams/99annual/abstracts/1360.htm (6.11), 1999. Seidl, R., Thom, D., Kautz, M., Martin-Benito, D., Peltoniemi, M., Vacchiano, G., Wild, J., Ascoli, D., Petr, M., Honkaniemi, J., Lexer" M.J., Trotsiuk, V., Mairota, P., Svoboda, M., Fabrika, M., Nagel T.A. and Reyer, C. P. O.: Forest disturbances under climate change, Nature Climate Change, 7(6), 395–402. doi:10.1038/nclim ate33 03, 2017. Suvanto, S., Henttonen, H. M., Nöjd, P., and Mäkinen, H.: Forest susceptibility to storm damage is affected by similar factors regardless of storm type: Comparison of thunder storms and autumn extra-tropical cyclones in Finland, Forest Ecology and Management, 381, 17‒28, doi: 10.1016/j.foreco.2016.09.005, 2016. Ulanova, N.G.: The effects of windthrow on forests at different spatial scales: a review. Forest Ecology and Management 135, 155–167. doi:10.1016/S0378-1127(00)00307-8, 2000.

Please also note the supplement to this comment:
https://essd.copernicus.org/preprints/essd-2020-91/essd-2020-91-AC1-supplement.pdf
* * *
Map figure with storm track and windthrow areas.

Legend:
- ---- Length of storm track
- —— Maximum width of storm track
- Windthrow areas
- 689 ID of windthrow area
- 480 ID of storm event
- Forest-covered area

Inset labels: 101603, 101601, 101596, 101598, 101594, 101590, 101587, 101590 EDAs with object ID

Windthrow area IDs: 686, 685, 684, 683, 681, 680, 679, 480

Scale: 0 10 20 km

Axis labels: 32° E, 33° E, 57°30' N, 57°

**Fig. 1.** Figure 6